# OVLR: Efficient, Scalable, and Robust Training via Output-Level Variance-Reduced Likelihood Ratio

**Minhao Zou** [1 2 3]   **Tao Ren** [4]   **Jinyang Jiang** [4]   **Rui Tao** [4]   **Zehao Li** [4]   **Jiale Fu** [5]   **Hui Shao** [6]   **Xianhua Liu** [1]   **Yijie Peng** [2 3 4 7 8 9]

## Abstract

Gradient-based optimization via backpropagation (BP) is inherently limited by the requirement of differentiability, rendering it inapplicable for piecewise-constant objectives with vanishing gradients (e.g., the hard 0-1 loss) or black-box feedback. While likelihood ratio (LR) methods offer a theoretical alternative, their high variance in high-dimensional spaces undermines training stability and scalability. We propose OVLR, a framework that makes direct optimization of gradient-agnostic objectives practical for modern deep networks by performing perturbations and antithetic sampling in the low-dimensional output space. OVLR achieves dramatic variance reduction while requiring only a single deterministic forward pass, with additional costs restricted to evaluating the loss function across multiple samples. On problems where BP provides gradients, OVLR remains competitive; on problems where BP fails to provide reliable learning signals, OVLR enables the direct optimization of objectives such as the 0-1 loss for noise-tolerant classification and truncated losses for outlier-resistant regression. Extensive empirical results across classification, generative modeling, language modeling, robot imitation learning, and black-box optimization confirm that OVLR is an effective tool for settings where standard gradient-based optimization is inapplicable. Code is available at https://github.com/MinhZou/OVLR.

[1]School of Computer Science, Peking University [2]Advanced Institute of Information Technology, Peking University [3]Hangzhou IndEngine Intelligence Co., Ltd. [4]Guanghua School of Management, Peking University [5]Dundee International Institute, Central South University [6]International Business School, Zhejiang University [7]Department of AI for Management, School of Management and Engineering, Nanjing University [8]Xiangjiang Laboratory [9]Wuhan Institute of Artificial Intelligence, Peking University. Correspondence to: Yijie Peng <pengyijie@pku.edu.cn>.

*Proceedings of the 43rd International Conference on Machine Learning*, Seoul, South Korea. PMLR 306, 2026. Copyright 2026 by the author(s).

## 1. Introduction

Gradient-based optimization methods form the cornerstone of modern deep learning (Ruder, 2016; Bengio et al., 2013). The backpropagation (BP) algorithm, which efficiently computes gradients via the chain rule, is the de facto standard for training deep networks. However, BP's effectiveness is confined to objectives that provide informative, deterministic gradients. This renders BP ineffective or inapplicable in two critical regimes: (1) objectives with vanishing or pathological gradients (e.g., the piecewise-constant 0-1 loss), and (2) black-box systems where gradients are inherently inaccessible. Consequently, many mathematically desirable gradient-agnostic objectives—such as robust losses for noise-tolerant classification and outlier-resistant regression—remain practically intractable under the standard BP paradigm.

To overcome these limitations, a range of gradient-free optimization methods has been developed. The score function estimator, also known as the likelihood ratio (LR) method (Glynn, 1990) or REINFORCE (Williams, 1992; Sutton et al., 1999), provides unbiased gradient estimates but suffers from high variance that severely hinders convergence. Evolutionary strategies (ES) (Salimans et al., 2017) and zeroth-order optimization (Nesterov & Spokoiny, 2017) estimate gradients through parameter space sampling; however, they become computationally prohibitive in high-dimensional spaces common in modern deep learning. Direct gradient approximation methods, such as the Straight-Through Estimator (Bengio et al., 2013), offer efficiency but introduce significant bias. Consequently, achieving a balance between gradient variance, computational scalability, and training stability remains a fundamental hurdle.

Among these, the Likelihood Ratio (LR) method has gained renewed interest for its flexibility and unbiasedness. Recent developments, such as the push-out likelihood ratio method (Peng et al., 2022), the unified likelihood ratio (ULR) approach (Jiang et al., 2024), and FLOPS (Ren et al., 2025a), have made significant progress in extending LR methods to neural network training without requiring explicit backpropagation through the loss. However, these methods often struggle with scalability when training large-scale neural networks. A primary reason is that injecting noise at the

input or hidden neuron level significantly increases gradient variance as model depth and parameter dimensionality grow, often restricting their applicability to smaller-scale or specialized settings.

Furthermore, while traditional variance reduction techniques—including control variates (Wilson, 1984), antithetic sampling (Hammersley & Morton, 1956), and multi-sampling (Burda et al., 2016) —can stabilize estimates, they typically demand excessive computational resources. In standard implementations, these methods often require multiple redundant forward passes through the entire computational graph, leading to a linear increase in training time that is prohibitive for modern deep learning pipelines. This creates a critical need for an LR framework that can achieve high-fidelity gradient estimation without compromising the efficiency of large-scale training.

In this work, we introduce OVLR (Output-Level Variance-Reduced Likelihood Ratio), a scalable framework that makes likelihood ratio methods practical for optimizing gradient-agnostic objectives in modern deep networks. The core innovation is shifting the perturbation and variance reduction mechanisms from the high-dimensional parameter space to the low-dimensional output space. By injecting structured symmetric noise at the model output and leveraging antithetic sampling, OVLR achieves a dramatic reduction in gradient variance.

From a computational standpoint, this design ensures that the resource-intensive forward pass through the network backbone is performed only once, while the evaluation of the loss function is repeated across multiple samples. Because the backbone's computation typically dominates the total training time, OVLR remains highly efficient and comparable to standard backpropagation. Furthermore, our framework integrates seamlessly into modern automatic differentiation pipelines via vector-Jacobian products without requiring architectural modifications. This allows OVLR to enable the direct optimization of gradient-agnostic objectives such as 0-1 loss for classification and truncated losses for regression, where standard backpropagation fails to provide reliable learning signals.

**Our contributions are summarized as follows:**

- **Scalable Variance Reduction**: OVLR achieves orders-of-magnitude variance reduction by performing perturbations and antithetic sampling in the low-dimensional output space. This design addresses the scalability limitation of likelihood ratio methods in high-dimensional parameter spaces, enabling stable training of deep architectures on gradient-agnostic objectives.

- **Efficient Autograd Integration**: Implemented via standard vector-Jacobian products, OVLR requires only a single forward pass per iteration while matching the per-step computational cost of backpropagation. This allows seamless integration as a practical alternative for loss backpropagation in gradient-agnostic settings within existing deep learning frameworks.

- **Robust Training**: We demonstrate that OVLR enables the direct optimization of gradient-agnostic objectives (e.g., 0-1 loss, truncated losses), establishing robust learning that significantly outperforms surrogate-based methods under label noise and outliers.

- **Extensive Validation**: We provide a comprehensive evaluation across computer vision, language modeling, robot imitation learning, and true black-box optimization, confirming that OVLR is competitive with standard backpropagation on problems with informative gradients, and enables effective optimization where gradients vanish or are inaccessible.

## 2. Related Work

Our work builds upon and extends research in gradient estimation, optimization in gradient-agnostic settings, and variance reduction. We position OVLR as a scalable and efficient method for optimizing gradient-agnostic objectives, complementing backpropagation in settings where differentiability requirements are not met.

### 2.1. Gradient Estimation via Likelihood Ratio Methods

Standard backpropagation (Rumelhart et al., 1986) is confined to objectives that provide informative gradients, failing when gradients vanish (e.g., piecewise-constant losses) or are inaccessible (e.g., black-box systems). Likelihood ratio (LR) methods, such as REINFORCE (Williams, 1992), provide unbiased gradient estimates but suffer from high variance. Recent advances like the push-out method (Peng et al., 2022), ULR (Jiang et al., 2024) and GLR (Peng et al., 2025) attempt to apply LR to neural networks but often perturb hidden neurons or intermediate layers, with applications in density estimation (Li et al., 2024; Li & Peng, 2025) and optimal transport (Chung et al., 2026). Unlike these approaches, OVLR restricts perturbations to the model's output space. This distinction is critical: preserving a deterministic forward path allows OVLR to leverage efficient Vector-Jacobian Products (VJPs) for the entire backbone. Consequently, our method achieves the scalability required for modern architectures like Transformers (Vaswani et al., 2017) and Mamba (Gu & Dao, 2024), avoiding the prohibitive variance and computational overhead inherent in methods that introduce stochasticity at the neuron level.

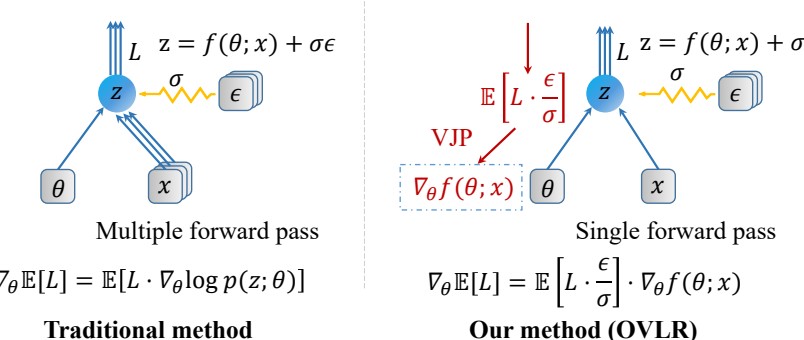

**Backpropagation form**

$L$   $z = f(\theta; x)$

BP   $\partial L/\partial z$

$\partial z/\partial \theta$   $\theta$   $x$

$$\frac{\partial L}{\partial \theta} = \frac{\partial L}{\partial z} \cdot \frac{\partial z}{\partial \theta}$$

**The chain rule**

**Likelihood Ratio form**

$L$   $z = f(\theta; x) + \sigma\epsilon$

$\sigma$   $\epsilon$

$z$

$\theta$   $x$

Multiple forward pass

$$\nabla_\theta \mathbb{E}[L] = \mathbb{E}[L \cdot \nabla_\theta \log p(z; \theta)]$$

**Traditional method**

$L$   $z = f(\theta; x) + \sigma\epsilon$

$\mathbb{E}\left[L \cdot \dfrac{\epsilon}{\sigma}\right]$   $\sigma$   $\epsilon$

VJP   $z$

$\nabla_\theta f(\theta; x)$   $\theta$   $x$

Single forward pass

$$\nabla_\theta \mathbb{E}[L] = \mathbb{E}\left[L \cdot \frac{\epsilon}{\sigma}\right] \cdot \nabla_\theta f(\theta; x)$$

**Our method (OVLR)**

*Figure 1.* Comparison of Gradient Estimation Paradigms. (Left) Standard Backpropagation relies on the deterministic chain rule and total differentiability of the entire graph. (Center) Traditional Likelihood Ratio methods (Score Function Estimators) often require multiple redundant forward passes, which limits scalability. (Right) The proposed OVLR preserves the efficiency of a single forward pass by decoupling the network backbone from output perturbations and implementing the estimator via Vector-Jacobian Products (VJP).

### 2.2. Optimizing Gradient-Agnostic Robust Objectives

Training with gradient-agnostic objectives (e.g., the hard 0-1 loss) typically relies on differentiable surrogate losses (Grabocka et al., 2019). However, these surrogates, such as Cross-Entropy, can be brittle under heavy label noise. Derivative-free methods like Evolutionary Strategies (ES) (Salimans et al., 2017) avoid the need for gradients but scale poorly with parameter dimensionality, becoming computationally prohibitive for modern deep architectures. OVLR addresses this gap by providing an unbiased and scalable pathway to directly optimize such robust objectives. By estimating gradients through efficient output-level sampling rather than high-dimensional parameter perturbations, OVLR enables robust and efficient training in settings where standard backpropagation fails and previous alternatives are intractable.

### 2.3. Variance Reduction and Efficiency

Variance reduction is essential for the practical utility of LR methods, and recent work has formalized the search for optimal gradient estimators by constructing variance minimization problems during training (Li et al., 2026; Ren et al., 2025b). While techniques like control variates (Wilson, 1984) and antithetic sampling (Hammersley & Morton, 1956) are well-established, their application to output-level perturbation in deep learning remains underexplored. Traditional multi-sample estimators (Burda et al., 2016) often require redundant forward passes through the entire network, incurring massive computational overhead. OVLR introduces output-level repetition and antithetic sampling specifically tailored to neural network computation graphs. By reusing shared deterministic components and integrating with vector-Jacobian products (VJPs), these techniques achieve substantial variance reduction with a total computa-

tional cost nearly identical to a single forward pass through the model backbone.

## 3. Unbiased Gradient Estimation via OVLR

We propose OVLR, which introduces perturbations at the model output and employs the likelihood ratio method to estimate gradients. This approach provides a gradient estimation framework primarily designed for gradient-agnostic objectives, while remaining applicable to standard differentiable losses. Figure 1 contrasts OVLR with standard backpropagation and traditional likelihood ratio methods, highlighting its efficiency and scalability enabled by output-level operations.

### 3.1. Problem Formulation

We consider a neural network parameterized by $\theta$ that produces a deterministic output $\mu(\theta) = f(\theta; x)$ for a given input $x$. Let $L(z, y)$ denote a general loss function, where $y$ is the ground truth label. In standard training paradigms, the objective is to minimize the expected loss over the data distribution $\mathcal{D}$: $L(\theta) = \mathbb{E}_{(x,y) \sim \mathcal{D}}[L(f(\theta; x), y)]$. Our goal is to minimize the expected loss $\mathbb{E}_z[L(z, y)]$ by estimating its gradient with respect to the network parameters $\theta$.

### 3.2. Gradient Estimation via Likelihood Ratio

To address scenarios where the loss function $L$ provides vanishing or no gradient signal with respect to the output, we adopt the likelihood ratio method by introducing controlled stochasticity into the model's output. Specifically, we add Gaussian noise to construct a stochastic output representation $z = \mu(\theta) + \sigma\epsilon, \quad \epsilon \sim \mathcal{N}(0, 1)$, where $\sigma > 0$ is a hyperparameter controlling the noise scale. Using the likelihood ratio trick, the gradient of the expected loss can

be estimated as:

$$\nabla_\theta \mathbb{E}_z[L(z,y)] = \mathbb{E}_{\epsilon \sim \mathcal{N}(0,1)}\left[L(z,y) \cdot \frac{\epsilon}{\sigma}\right] \cdot \nabla_\theta \mu(\theta). \quad (1)$$

This formulation yields an unbiased gradient estimator that remains valid even when $L$ is a gradient-agnostic function of $z$, making it broadly applicable. Note that OVLR estimates the loss gradient at the output level and propagates this signal through the model via a single Vector-Jacobian Product (VJP), treating the loss $L(z,y)$ itself as a black-box function without requiring gradient information from it. This distinguishes OVLR from pure likelihood ratio methods that operate at the parameter level, and from standard backpropagation which requires a differentiable computational graph. The detailed derivation and theoretical analysis of OVLR are provided in Appendices A, B, and C.

Let $g(\epsilon) = L(\mu + \sigma\epsilon, y) \cdot \frac{\epsilon}{\sigma}$ represent the stochastic output-level gradient component. The Monte Carlo estimate of the output-level gradient signal over $n$ independent samples is then defined as:$\hat{g}_{MC} = \frac{1}{n}\sum_{i=1}^{n} g(\epsilon_i), \quad \epsilon_i \sim \mathcal{N}(0,1)$, where each $\epsilon_i$ is sampled from a standard normal distribution. This estimated signal $\hat{g}_{MC}$ is unbiased but may suffer from high variance, particularly when the loss function $L$ provides little gradient information or the noise scale $\sigma$ is large.

### 3.3. Variance Reduction Methods

#### 3.3.1. OUTPUT-LEVEL REPETITION

To improve the statistical efficiency of our gradient estimator, we introduce a variance reduction strategy based on output-level repetition, which is both more effective and computationally efficient compared to conventional input-level repetition. In our model, the deterministic mean $\mu(\theta)$ is shared across all samples, allowing it to be computed once and reused, which eliminates redundant forward computations. This results in $\hat{g}_{out} = \frac{1}{n}\sum_{i=1}^{n} L(\mu + \sigma\epsilon_i, y) \cdot \frac{\epsilon_i}{\sigma}$. The variance of output-level gradient signal is $\text{Var}[\hat{g}_{out}] = \frac{1}{n}\text{Var}\left[L(z,y) \cdot \frac{\epsilon}{\sigma}\right]$. In traditional input-level repetition, the model forward pass is repeatedly computed for identical inputs under different noise realizations, leading to substantial computational overhead. In contrast, output-level repetition performs the stochastic operations after a single deterministic forward pass, thereby significantly reducing redundant computation while preserving gradient signal diversity.

#### 3.3.2. ANTITHETIC SAMPLING ON OUTPUT-LEVEL REPEATED SAMPLES

Building upon output-level repeated samples, we further reduce variance by employing an antithetic sampling strategy. For $n$ even, we generate $\epsilon_1, \ldots, \epsilon_{n/2} \sim \mathcal{N}(0,1)$ and set $\epsilon_{i+n/2} = -\epsilon_i, \quad \text{for } i = 1, \ldots, n/2$. Under this formulation, the antithetic output-level gradient signal $\hat{g}_{anti}$ is

expressed as: $\hat{g}_{anti} = \frac{1}{n}\sum_{i=1}^{n/2}[g(\epsilon_i) + g(-\epsilon_i)]$. This estimator remains unbiased due to the symmetry of the Gaussian distribution and the linear influence of the noise term $\epsilon$ in the gradient calculation. Furthermore, under mild regularity conditions, this strategy ensures that $\text{Var}[\hat{g}_{anti}] \leq \text{Var}[\hat{g}_{out}]$. In particular, if the stochastic term $g(\epsilon)$ behaves as an approximately odd function—where $g(-\epsilon) \approx -g(\epsilon)$—the summation $g(\epsilon) + g(-\epsilon)$ concentrates near zero. This results in dominant noise cancellation and a substantial reduction in the overall variance of the gradient estimates

### 3.4. Algorithm Description

We begin with a general-purpose likelihood ratio gradient estimation algorithm applicable to both standard differentiable losses and gradient-agnostic objectives in supervised learning. Our vectorized approach, which efficiently handles batch processing and antithetic sampling, is detailed in Algorithm 1.

---

**Algorithm 1** Vectorized Training with OVLR

1: **Input:** Mini-batch $(X, Y)$, parameters $\theta$, noise scale $\sigma$, sample count $n$, batch size $B$, feature dimension $d$
2: **Output:** Updated model parameters $\theta'$
3: **Forward Pass:**
4: $Z = f(\theta; X)$ {Single deterministic backbone pass; $Z \in \mathbb{R}^{B \times d}$}
5: $\tilde{Z} = Z.\text{repeat}(n, 1), \tilde{Y} = Y.\text{repeat}(n)$ {Vectorized expansion to $N = B \times n$ samples}
6: **Symmetric Sampling:**
7: Sample $\epsilon_{1:N/2} \sim \mathcal{N}(0, I_d)$, set $\epsilon_{N/2+1:N} = -\epsilon_{1:N/2}$ {Antithetic sampling}
8: $Z_{perturbed} = \tilde{Z} + \sigma\epsilon$ {Batch-wise perturbation of features with Gaussian noise}
9: **Loss & Gradient Construction:**
10: $L = l(Z_{perturbed}, \tilde{Y})$ {Batch-wise black-box loss evaluation}
11: $v = \frac{1}{N\sigma}(L \odot \epsilon)$ {Vectorized output-level gradient signal; $\odot$ is element-wise product}
12: **Backward Pass:**
13: $\nabla_\theta L = \text{VJP}(Z, \theta, v)$ {Single VJP through backbone network (equivalent to $Z.\text{backward}(v)$ in PyTorch)}
14: $\theta' = \text{Optimizer}(\theta, \nabla_\theta L)$ {Update parameters using optimizer}
15: **Return** $\theta'$

---

## 4. Computational Complexity and Efficiency

OVLR exhibits superior scalability by decoupling the resource-intensive model backbone from the stochastic estimation process, as summarized in Table 1. Regarding computational throughput, standard backpropagation (BP) requires symmetric forward and backward passes through

both the model and the loss function. While traditional input-level sampling methods scale model-level costs $(T_\mu, \tilde{T}_\mu)$ linearly with the sample size $n$, OVLR maintains a constant backbone cost. It executes only a single deterministic forward pass and a single vector-Jacobian product (VJP), regardless of the number of samples. As illustrated in Figure 1, this efficiency is achieved by confining the $n$-fold increase in computation to the lightweight loss evaluations $(n \cdot T_L)$ and the signal construction phase. In modern deep architectures, these operations are several orders of magnitude cheaper than the backbone passes, ensuring that OVLR maintains high training throughput even with large sample sizes.

From a memory perspective, OVLR achieves high resource efficiency by bypassing redundant graph caching and high-dimensional activation storage. By maintaining a deterministic forward path, the system caches only a single set of activations $(M_\mu)$ to support the VJP, whereas input-level methods must store $n$ distinct sets, which frequently leads to memory exhaustion in deep networks. Crucially, by treating the loss function as a black box, OVLR enables a graph-free evaluation that eliminates the memory footprint of the loss function's computational graph $(M_L \to 0)$. This is particularly advantageous for complex or gradient-agnostic objectives that are otherwise prohibitive for standard BP. While the memory required for noise samples $(M_{\text{noise}})$ scales with $n$, this overhead remains marginal as the stochasticity is restricted to the low-dimensional output space. Consequently, the peak memory usage of OVLR remains nearly identical to standard BP, facilitating its scalability to large-scale, high-parameter models.

## 5. Experimental Evaluation

This section presents a systematic empirical evaluation of the OVLR framework across diverse tasks and architectures. Our investigation aims to validate: (i) the efficacy of combined sampling strategies in variance reduction; (ii) the competitiveness of OVLR against standard backpropagation (BP) on problems with informative gradients; and (iii) its unique capability in optimizing gradient-agnostic objectives and robust regression landscapes. More detailed results across generative modeling, language modeling, and robot imitation learning are presented in Appendix E, Appendix F, and Appendix G for completeness.

### 5.1. Efficacy of Variance Reduction Strategies

We evaluate four variance reduction configurations—Vanilla Perturbation, Antithetic Sampling (AS), Output-Level Repetition, and their combination—using a CNN (LeCun et al., 1998) backbone on CIFAR-10 (Krizhevsky, 2009). We sweep the noise scale $\sigma \in [0.01, 1.0]$ across 50 values to analyze gradient stability. Detailed architecture, training

configuration, and noise sampling procedures are provided in Appendix L.

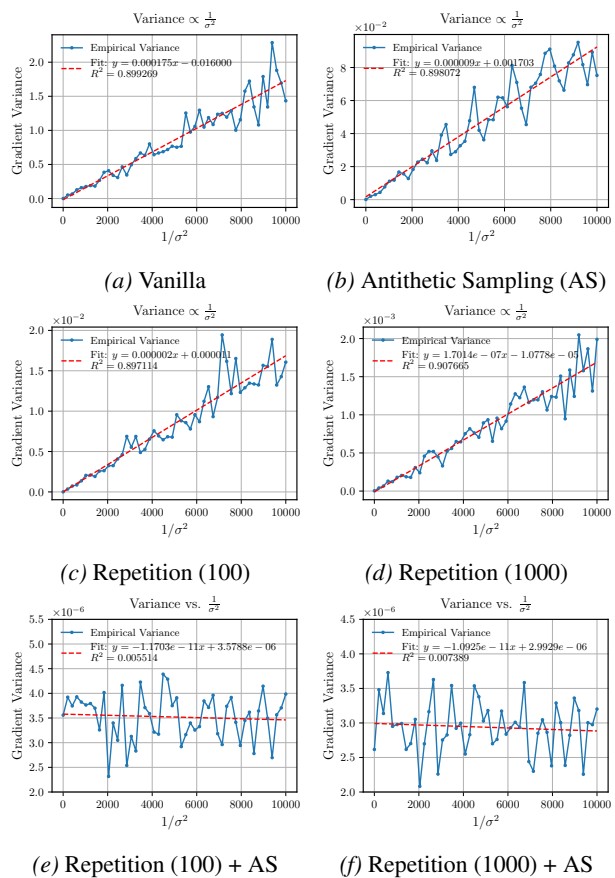

*(a)* Vanilla     *(b)* Antithetic Sampling (AS)

*(c)* Repetition (100)     *(d)* Repetition (1000)

*(e)* Repetition (100) + AS     *(f)* Repetition (1000) + AS

*Figure 2.* Variance Reduction Efficacy of Different Strategies. Gradient variance as a function of inverse noise squared ($1/\sigma^2$) for vanilla perturbation, antithetic sampling (AS), output-level repetition, and their combination. The combined strategy (AS + repetition) effectively neutralizes the dominant $1/\sigma^2$-dependent noise, yielding near-constant variance.

**Variance Stabilization Analysis.** As illustrated in Figure 2, the vanilla estimator exhibits a variance that grows linearly with $1/\sigma^2$, confirming its high sensitivity to noise. In contrast, applying AS (Figure 2b) or 100-fold repetition (Figure 2c) suppresses variance by approximately two orders of magnitude. The most significant stabilization is achieved by the combined strategy (Figure 2e-2f), where the fitted variance curve remains near-constant and independent of $1/\sigma^2$ ($R^2 \approx 0$). This confirms that our framework effectively neutralizes dominant noise sources through symmetric cancellation and repetition.

**Ablation on Practical Performance.** Table 2 provides an ablation study using a ResNet-18 backbone with cross-entropy loss, evaluating model accuracy on CIFAR-100 (100 classes) (Krizhevsky, 2009) and TinyImageNet (200 classes). AS consistently boosts Top-1 performance, while

*Table 1.* Computational and memory profiles during training. $T_\mu/\tilde{T}_\mu$ and $T_L$ denote the costs of a single model pass and loss evaluation, respectively. $M_\mu$ and $M_\theta$ represent memory for backbone activations and parameters. $M_{\text{noise}}$ denotes the memory for storing output-level perturbations, which scales with $n$ but remains marginal due to its low dimensionality. Notably, OVLR treats the loss as a black box, eliminating the need to cache its computational graph ($M_L \to 0$) and decoupling the resource-intensive backbone from the sample size $n$.

| Method | Forward Cost | Backward Cost | Memory Usage |
|---|---|---|---|
| Standard Backpropagation | $T_\mu + T_L$ | $\tilde{T}_\mu + \tilde{T}_L$ | $M_\mu + M_L + M_\theta$ |
| Input-level Repetition (Jiang et al., 2024; Ren et al., 2025a) | $n \cdot T_\mu + n \cdot T_L$ | $n \cdot \tilde{T}_\mu + T_{\text{vec}}$ | $n \cdot M_\mu + n \cdot M_{\text{noise}} + M_\theta$ |
| **OVLR (Ours)** | $T_\mu + n \cdot T_L$ | $\tilde{T}_\mu + T_{\text{vec}}$ | $M_\mu + n \cdot M_{\text{noise}} + M_\theta$ |

*Table 2.* Ablation Study on Variance Reduction Strategies and Noise Robustness. Top-1 test accuracy (%) on CIFAR and TinyImageNet using ResNet-18. The table compares vanilla vs. antithetic sampling (AS), explores the effect of repetition count ($n$), and demonstrates the model's insensitivity to a wide range of noise scales ($\sigma$) when AS is combined with sufficient repetition ($n \geq 200$)

| Dataset | Mode Vanilla | AS | Repetition ($n$) 1 | 20 | 50 | 100 | 200 | 300 | 400 | 500 | Noise Scale $\sigma$ 0.01 | 0.1 | 0.5 | 1 | 2 | 5 |
|---|---|---|---|---|---|---|---|---|---|---|---|---|---|---|---|---|
| CIFAR10 | 88.15 | 88.47 | 30.27 | 86.86 | 88.06 | 88.16 | 88.47 | 88.34 | 87.35 | 88.40 | 87.96 | 88.33 | 87.21 | 88.47 | 87.76 | 88.20 |
| CIFAR100 | 36.62 | 60.06 | 1.85 | 38.43 | 54.82 | 59.77 | 60.06 | 61.20 | 61.73 | 62.45 | 60.94 | 59.32 | 59.92 | 60.06 | 60.26 | 58.98 |
| TinyImageNet | 18.78 | 46.90 | 0.71 | 22.05 | 37.91 | 44.94 | 46.90 | 45.68 | 45.69 | 46.39 | 46.92 | 46.36 | 45.24 | 46.90 | 45.99 | 47.31 |

output-level repetition provides steady gains that saturate around $n = 200$. Crucially, the combination of these techniques renders the training process largely insensitive to the specific choice of $\sigma \in [0.1, 5.0]$, facilitating robust hyperparameter selection in practical deployment.

### 5.2. Scalability and Performance with Informative Gradients

To verify that OVLR remains competitive on standard differentiable tasks, we first benchmark it in the setting where backpropagation excels. We benchmark it against BP across ResNet-18 (He et al., 2016), DenseNet-121 (Huang et al., 2017), and ViT-B/16 (Dosovitskiy et al., 2021) on CIFAR-100 (Krizhevsky, 2009) and TinyImageNet datasets. Additional experimental details are provided in Appendix L.

*Table 3.* OVLR Achieves Competitive Accuracy with Negligible Overhead. Comparison of test accuracy (%) and training time per epoch (s) between standard backpropagation (BP) and our OVLR method across architectures and datasets. OVLR maintains accuracy within 1% of BP while introducing minimal runtime increase (often ¡ 0.5%), confirming its efficiency.

| Model | CIFAR100 Accuracy (BP/OVLR) | Time (s) (BP/OVLR) | TinyImageNet Accuracy (BP/OVLR) | Time (s) (BP/OVLR) |
|---|---|---|---|---|
| ResNet-18 | 63.10 / 62.10 -1.00 | 48.85 / 49.19 +0.34 | 47.27 / 46.83 -0.44 | 97.87 / 98.22 +0.35 |
| DenseNet-121 | 65.92 / 65.31 -0.61 | 153.62 / 153.53 -0.09 | 53.88 / 52.86 -1.02 | 307.10 / 306.92 -0.18 |
| ViT-B/16 | 86.79 / 88.17 +1.38 | 373.06 / 373.75 +0.69 | 79.72 / 79.89 +0.17 | 746.74 / 748.05 +1.31 |

**Accuracy and Training Throughput.** As summarized in Table 3, OVLR achieves performance competitive with BP, typically maintaining a Top-1 accuracy margin within 1%. Notably, OVLR outperforms BP on the ViT-B/16 architecture for CIFAR-100 (88.17% vs. 86.79%). In terms of temporal efficiency, the training time per epoch remains nearly identical to standard BP across all models. This confirms that the $n$ lightweight loss evaluations in OVLR introduce negligible overhead compared to the model's primary forward-backward pass. The nearly identical convergence trajectories of BP and OVLR, shown in Figure 3, further validate that our method does not compromise training stability or speed while matching final performance.

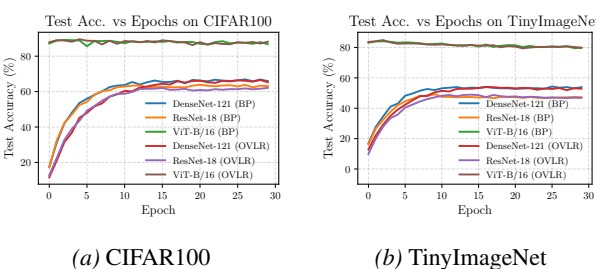

*(a)* CIFAR100      *(b)* TinyImageNet

*Figure 3.* Convergence Trajectories of BP and OVLR Are Nearly Identical. Test accuracy curves across training epochs on CIFAR-100 and TinyImageNet show that OVLR closely matches the convergence speed and final performance of standard backpropagation on problems with informative gradients, establishing its competitiveness in the standard supervised setting.

**Memory Footprint and Optimization.** Table 5 highlights the memory characteristics of our framework. While OVLR introduces a modest increase in allocated memory

for noise storage (e.g., +60MB for ViT on TinyImageNet), it effectively optimizes reserved memory. Specifically, for ResNet-18 on TinyImageNet, we observe a reduction in reserved memory (-14MB). This is attributed to our graph-free evaluation: by treating the loss as a black-box, OVLR eliminates the need to cache the computational graph for the loss function ($M_L \to 0$), allowing the framework to optimize memory reuse more aggressively than standard BP.

**Comparison with LR-Family Methods.** To isolate the contribution of perturbation location, we compare OVLR against LR-family methods on MNIST with the same backbone (SimpleCNN) and cross-entropy loss. As shown in Table 4, OVLR achieves near-BP accuracy (96.80%) with matching time and memory, while vanilla LR with parameter perturbation suffers prohibitive cost (428.74s, 4835MB). GLR-style input-level repetition improves accuracy but remains $2.6\times$ slower and uses $1.6\times$ more memory. The $14\times$ speedup and $73\times$ memory reduction of OVLR over parameter-space LR stems from output-level perturbation design, confirming the theoretical analysis in Section 4.

*Table 4.* Comparison with LR-Family Methods on MNIST (SimpleCNN, Cross-Entropy Loss). OVLR achieves near-BP performance with minimal overhead, while vanilla LR methods suffer from prohibitive cost.

| Method | Accuracy (%) | Time (s) | Memory (MB) |
|---|---|---|---|
| BP | 97.35 | 29.82 | 65.55 |
| **OVLR** | 96.80 | 30.02 | 65.55 |
| GLR-style (input-level) | 94.60 | 76.98 | 107 |
| Vanilla LR (param-level) | 82.70 | 428.74 | 4835 |

**Scalability with Output Dimensionality.** We conduct a controlled study varying the number of classes $C$ from 10 to 5000 with increasing repetition budgets. With repeat=2000, the gap between OVLR and BP shrinks to 0.17% at $C$=500, 0.32% at $C$=1000, 0.95% at $C$=2000, and 1.81% at $C$=5000. Higher output dimensions demand larger repetition budgets, but the gap closes systematically. Full results are provided in Appendix H.

### 5.3. Robust Classification: Direct Optimization Gradient-Agnostic Objectives

We evaluate OVLR's capability in scenarios where analytic gradients are unavailable or uninformative, specifically focusing on the Hard 0-1 loss, $L_{0\text{-}1}(\mathbf{z}, y) = \mathbb{I}(\arg\max_k z_k \neq y)$, where $\mathbf{z} \in \mathbb{R}^K$ denotes the model logits, $y$ is the true label, and $\mathbb{I}(\cdot)$ is the indicator function, for image classification tasks.

**Overcoming BP Failure in Piecewise-Constant Objectives.** Table 6 demonstrates that standard BP fails to provide any effective learning signal for the gradient-agnostic

*Table 5.* Memory Profile: OVLR Optimizes Reserved Memory via Graph-Free Evaluation. Comparison of peak GPU memory (allocated vs. reserved) shows that while OVLR adds a small, fixed overhead for noise storage (e.g., +20–60 MB in allocated), its graph-free design eliminates loss graph caching ($M_L \to 0$), enabling more aggressive memory reuse and sometimes reducing reserved memory (e.g., -14 MB for ResNet-18/TinyImageNet)

| Model | CIFAR100 | | TinyImageNet | |
|---|---|---|---|---|
| | Allocated (BP/OVLR) | Reserved (BP/OVLR) | Allocated (BP/OVLR) | Reserved (BP/OVLR) |
| ResNet-18 | 1,565 / 1,587 | 3,466 / 3,466 | 1,565 / 1,625 | 3,468 / 3,454 |
| | +22 | 0 | +60 | -14 |
| DenseNet-121 | 8,112 / 8,132 | 9,334 / 9,338 | 8,113 / 8,174 | 9,348 / 9,418 |
| | +20 | +4 | +61 | +70 |
| ViT-B/16 | 10,082 / 10,102 | 10,394 / 10,406 | 10,083 / 10,144 | 10,402 / 10,504 |
| | +20 | +12 | +61 | +102 |

Hard 0-1 loss, as its gradient is zero almost everywhere. In contrast, OVLR successfully optimizes this loss by perturbing the model outputs and constructing a gradient estimate via variance-reduced finite differences, reaching 61.74% Top-1 accuracy on CIFAR-10. While standard BP fails on 0-1 loss, we also compare against Mixed Loss to demonstrate OVLR's ability to optimize the exact objective directly. By utilizing a Mixed Loss ($\alpha = 0.5$), our framework bridges the gap between the evaluation metric and a tractable optimization objective, reaching performance levels (87.92% on CIFAR-10) that are competitive with standard cross-entropy training.

*Table 6.* Optimization Performance on Losses with Informative vs. Gradient-Agnostic Signals. Performance on the Hard 0-1 Loss is reported as (Standard BP / OVLR). Standard BP fails to provide learning signals due to the vanishing gradient of the objective.

| Dataset | Standard Cross-Entropy | | | Hard 0-1 Loss (BP / OVLR) | | | Mixed Loss ($\alpha = 0.5$) | | |
|---|---|---|---|---|---|---|---|---|---|
| | Top1 | Top3 | Top5 | Top1 | Top3 | Top5 | Top1 | Top3 | Top5 |
| CIFAR10 | 87.72 | 97.89 | 99.50 | – / 61.74 | – / 76.47 | – / 79.39 | 87.92 | 97.81 | 99.54 |
| CIFAR100 | 62.11 | 80.90 | 86.69 | – / 5.25 | – / 7.45 | – / 9.19 | 62.80 | 80.19 | 86.41 |

**Comparison with Direct Estimators on Hard 0-1 Loss.** We further compare OVLR against methods that directly optimize the hard 0-1 loss. As shown in Table 7, among direct estimators, OVLR achieves the best or tied-best accuracy while being the fastest. PPO (Schulman et al., 2017) ties on CIFAR-10 but requires 65% more training time. Zeroth-order methods (Nesterov & Spokoiny, 2017) fail to scale. STE-Style (Bengio et al., 2013) and Surrogate methods achieve higher accuracy but do not directly optimize the 0-1 objective, relying on cross-entropy proxies instead.

**Robustness to Label Noise.** We further evaluate the Hard 0-1 objective under symmetric label noise (rates $r \in [0.1, 0.6]$). As shown in Figure 4 and Table 8, standard cross-entropy (CE) suffers from severe noise memorization, leading to a sharp decline in test accuracy. Conversely, optimizing the gradient-agnostic 0-1 objective via OVLR proves inherently more robust, as it is defined solely by

*Table 7.* Direct estimators on hard 0-1 loss. Accuracy (%) and time (s). [†]Optimizes CE proxy, not 0-1 directly.

| Method | MNIST | Time | CIFAR10 | Time |
|---|---|---|---|---|
| **OVLR** | 95.65 | 28.6 | 62.04 | 225.1 |
| REINFORCE (Williams, 1992) | 95.50 | 44.5 | 56.02 | 295.6 |
| PPO (Schulman et al., 2017) | 95.30 | 72.4 | 62.12 | 373.2 |
| GRPO (Shao et al., 2024) | 94.80 | 66.7 | 45.26 | 374.4 |
| Zeroth-Order (Nesterov & Spokoiny, 2017) | 70.35 | 45.0 | 23.92 | 292.6 |
| STE-Style[†] (Bengio et al., 2013) | 95.95 | 32.4 | 67.66 | 223.8 |
| Surrogate[†] | 96.10 | 29.8 | 68.54 | 211.1 |

the final prediction correctness and is thus less sensitive to the misleading supervisory signals from noisy labels. At $r = 0.6$, OVLR achieves a peak accuracy of 73.0% (a +6.7% improvement over CE), demonstrating superior stability and generalization in corrupted supervised settings. Note that the OVLR framework utilizes a 5-epoch warmup with Cross-Entropy (CE) before transitioning to the 0-1 loss optimization phase to ensure stable initial convergence.

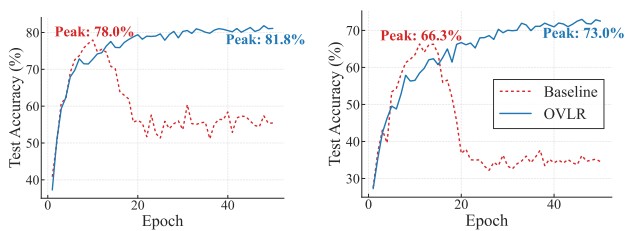

*(a)* Under 40% Label Noise    *(b)* Under 60% Label Noise

*Figure 4.* OVLR Filters Noisy Gradients for Robust Convergence. Test accuracy curves on CIFAR-10 under 40% and 60% symmetric label noise. While standard training degrades, OVLR maintains stable convergence and higher final accuracy by estimating gradients that are resilient to corrupt labels, effectively rejecting noise-induced update directions.

*Table 8.* OVLR Exhibits Enhanced Robustness Under High Label Noise. Peak test accuracy (%) on CIFAR-10 under varying symmetric noise rates (0.1–0.6). The improvement over the standard cross-entropy (CE) baseline grows as the noise rate increases (from +2.4% to +6.7%), with the best results for each noise rate highlighted in bold.

| METHOD | 0.1 | 0.2 | 0.3 | 0.4 | 0.5 | 0.6 |
|---|---|---|---|---|---|---|
| BASELINE (CE) | 83.8 | 81.4 | 79.4 | 78.0 | 72.6 | 66.3 |
| PROPOSED (OVLR) | 86.2 | 85.5 | 83.9 | 81.8 | 78.8 | 73.0 |
| IMPROVEMENT ($\Delta$) | +2.4 | +4.1 | +4.5 | +3.8 | +6.2 | +6.7 |

## 5.4. Robust Regression: High-Fidelity Signal Recovery under Extreme Outliers

To evaluate the robustness of OVLR under extreme outliers, we employ the Truncated $L_1$ loss, $L_{\text{Trunc}}(y, \hat{y}; \tau) = \min(|y - \hat{y}|, \tau)$, which bounds the influence of any single sample by ignoring errors larger than a threshold $\tau$. We test this on a one-dimensional sine wave regression task

with $N_{\text{train}} = 400$ samples, where a fraction $r = 20\%$ are corrupted by a constant shift of $\delta = +10.0$. Our method, OVLR$_{\text{Trunc}}$, is compared against three baselines optimized via standard backpropagation: BP$_{\text{MSE}}$, BP$_{\ell_1}$, and BP$_{\text{Trunc}}$. All models are multi-layer perceptrons trained with Adam (lr = 0.01) for 2000 epochs. To systematically assess hyperparameter sensitivity, we conduct a grid search over noise scale $\sigma$ and truncation threshold $\tau$ (both in $\{0.1, 0.5, 1.0, 2.0, 5.0\}$) across three outlier scenarios.

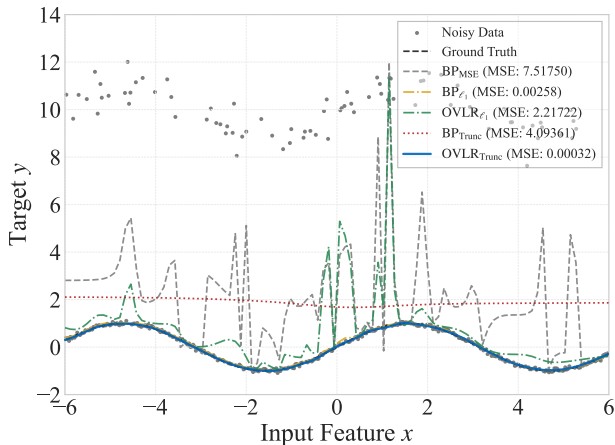

*Figure 5.* OVLR Overcomes Gradient Vanishing in Truncated Loss Optimization. Regression under 20% outliers ($\delta = +10.0$). While BP$_{\text{Trunc}}$ fails due to zero gradients in the truncated region (flat loss surface), OVLR$_{\text{Trunc}}$ successfully recovers the ground-truth signal by providing gradient estimates via output perturbations.

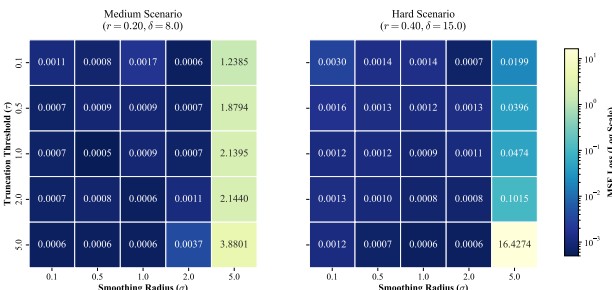

*Figure 6.* Broad Hyperparameter Robustness Simplifies Deployment. Sensitivity of final MSE to noise scale $\sigma$ and truncation threshold $\tau$ across outlier scenarios. The extensive plateaus of low error (blue) indicate that OVLR's performance is stable over a wide range of $(\sigma, \tau)$ pairs, reducing the need for precise hyperparameter tuning.

**Superior Signal Recovery and Gradient Stability.** The results, shown in Figure 5, reveal a clear advantage for our approach. While BP$_{\text{MSE}}$ is severely skewed by outliers (MSE: 7.51750) and BP$_{\text{Trunc}}$ fails to recover the underlying signal due to vanishing gradients in the truncated region (MSE: 4.09361), OVLR$_{\text{Trunc}}$ accurately reconstructs the ground-truth sine wave with minimal error (MSE: 0.00032).

This success stems from OVLR's ability to provide meaningful gradient estimates even in the flat, zero-gradient regions of the truncated loss, overcoming a fundamental limitation of gradient-based optimization for such non-convex, truncated objectives.

**Broad Hyperparameter Robustness.** Furthermore, the sensitivity analysis in Figure 6 demonstrates that OVLR performs robustly across a broad range of $(\sigma, \tau)$ pairs. In the most challenging setting with $40\%$ outliers, a moderate smoothing radius ($\sigma \in [0.5, 2.0]$) yields near-optimal MSE ($\approx 0.0006$), whereas excessive smoothing ($\sigma = 5.0$) causes the estimator to be biased by the outlier cluster, increasing the error to 16.4274. This indicates that while the method is stable, selecting $\sigma$ within a reasonable middle range is sufficient for optimal performance, simplifying practical deployment.

### 5.5. True Black-Box Optimization

To validate OVLR in genuinely gradient-free settings, we evaluate on IOHexperimenter benchmarks (de Nobel et al., 2024) where the optimizer receives only scalar objective values. As shown in Table 9, on discrete PBO benchmarks, OVLR achieves a mean normalized score of 0.9917 with $86.7\%$ success rate, compared to REINFORCE's 0.7040 with $0\%$ success. On continuous BBOB benchmarks, OVLR achieves 0.2389 normalized score with $25\%$ target-hit rate, outperforming CEM (Rubinstein, 1999) (0.1922) and Gaussian-REINFORCE (0.0431). Full per-problem results are in Appendix K.

*Table 9.* True black-box optimization on IOH benchmarks.

| Suite | Method | Norm. Score | Success Rate |
|---|---|---|---|
| Discrete PBO | **OVLR** | **0.9917** | **86.7%** |
| | (1+1)-EA | 0.9688 | 80.0% |
| | REINFORCE | 0.7040 | 0.0% |
| Cont. BBOB | **OVLR** | **0.2389** | **25.0%** |
| | CEM | 0.1922 | 25.0% |
| | Gauss-REINF. | 0.0431 | 0.0% |

### 6. Limitations

The OVLR framework, while designed for efficiency, presents several considerations for future scaling. First, its gradient estimator requires multiple evaluations of the loss function, which may become a computational bottleneck if the loss itself is exceptionally expensive to compute. This is particularly relevant for gradient-agnostic losses that lack closed-form expressions. Second, for tasks with extremely high-dimensional outputs, the simultaneous storage of multiple perturbed output vectors could increase memory overhead; moreover, our output dimension scaling study shows

that very high-dimensional outputs (e.g., $C=5000$) require larger repetition budgets to close the gap with BP. Third, our theoretical convergence guarantees rely on bounded-loss assumptions (Assumptions B.1–B.2), which may not hold in settings with unbounded losses such as certain reinforcement learning problems; extending the theoretical framework to unbounded-loss regimes requires additional technical conditions beyond the scope of this work. These aspects point to productive avenues for further optimization in large-scale or specialized applications.

### 7. Conclusion

In this work, we introduced the OVLR framework, a scalable solution for gradient estimation that makes likelihood ratio methods practical for optimizing gradient-agnostic objectives in modern deep networks. By shifting stochasticity to the low-dimensional output space and leveraging VJPs, OVLR achieves gradient estimation that is both variance-stable and computationally decoupled from the backbone complexity. Our theoretical analysis and extensive empirical results demonstrate that OVLR is competitive with standard backpropagation on problems with informative gradients, while requiring only a single deterministic forward pass. Crucially, OVLR enables the direct and scalable optimization of gradient-agnostic objectives. We demonstrated its efficacy on the hard 0-1 loss for noise-tolerant classification, truncated losses for outlier-resistant regression, and true black-box optimization—scenarios where standard backpropagation fails due to vanishing or inaccessible gradients. OVLR is not intended as a universal replacement for backpropagation; rather, it is a specialized tool for the important regime where BP fails, providing a practical path for robust learning and optimization of objectives where standard gradient-based methods are inapplicable.

### Impact Statement

This paper presents work whose goal is to advance the field of machine learning. There are many potential societal consequences of our work, none of which we feel must be specifically highlighted here.

### Acknowledgments

This work was supported in part by the National Natural Science Foundation of China under Grants 72325007 and 72250065, the Key Project of Xiangjiang Laboratory under Grant 23XJ02004, the Major Project of Xiangjiang Laboratory under Grant 24XJJCYJ01001, and the Science and Technology Innovation Program of Hunan Province under Grant 2024RC7003.

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

## A. Proof and Derivation of the OVLR Gradient Estimator

### A.1. Case I: Loss Function with Informative Gradients

#### A.1.1. ASSUMPTIONS

We make the following assumptions for the derivation:

- The loss function $L(z, y)$ is differentiable with respect to $z$ (i.e., it provides an informative gradient $\frac{\partial L}{\partial z}$), and both $L(z, y)$ and its derivative decay sufficiently fast as $z \to \pm\infty$, so that integration by parts is valid and boundary terms vanish.

- The label $y$ is observed and fixed, and does not depend on the random variable $\epsilon$. Therefore, the loss function $L(z, y)$ can be written as $L(z)$ without loss of generality during expectation over $\epsilon$.

- The mapping $z = \mu + \sigma\epsilon$, where $\epsilon \sim \mathcal{N}(0, 1)$, defines a reparameterization of the random variable $z$.

We aim to estimate the gradient of the expected loss with respect to the model parameters $\theta$ when the loss function $L(z)$ provides an informative gradient with respect to its input $z$. The model output is given by:

$$z = f(\theta; x) + \sigma\epsilon, \quad \epsilon \sim \mathcal{N}(0, 1) \tag{2}$$

Let $\mu(\theta) = f(\theta; x)$. Then $z = \mu(\theta) + \sigma\epsilon$.

We are interested in estimating:

$$\nabla_\theta \mathbb{E}_{\epsilon \sim \mathcal{N}(0,1)}[L(z)] = \nabla_\theta \mathbb{E}_\epsilon[L(\mu(\theta) + \sigma\epsilon)] \tag{3}$$

Under suitable regularity conditions, we can interchange the gradient and expectation:

$$\nabla_\theta \mathbb{E}_\epsilon[L(\mu(\theta) + \sigma\epsilon)] = \mathbb{E}_\epsilon[\nabla_\theta L(\mu(\theta) + \sigma\epsilon)] \tag{4}$$

Because $L$ is differentiable with respect to $z$, and $z$ is a function of $\theta$ through $\mu(\theta)$, we can apply the chain rule:

$$\nabla_\theta L(z) = \nabla_\theta L(\mu(\theta) + \sigma\epsilon) = \frac{\partial L}{\partial z} \cdot \frac{\partial z}{\partial \theta} = \frac{\partial L}{\partial z} \cdot \nabla_\theta \mu(\theta) \tag{5}$$

Therefore, since $\nabla_\theta \mu(\theta)$ is independent of the noise $\epsilon$, it can be moved outside the expectation:

$$\nabla_\theta \mathbb{E}_\epsilon[L(z)] = \mathbb{E}_\epsilon\left[\frac{\partial L}{\partial z} \cdot \nabla_\theta \mu(\theta)\right] = \mathbb{E}_\epsilon\left[\frac{\partial L}{\partial z}\right] \cdot \nabla_\theta \mu(\theta) \tag{6}$$

To simplify $\mathbb{E}_\epsilon\left[\frac{\partial L}{\partial z}\right]$, we use the reparameterization trick. Let:

$$\epsilon = \frac{z - \mu(\theta)}{\sigma} \tag{7}$$

From the proof via Stein's Lemma (A.1.2) or integration by parts (A.1.3), we know:

$$\mathbb{E}_\epsilon\left[L(z) \cdot \frac{\epsilon}{\sigma}\right] = \mathbb{E}_\epsilon\left[\frac{\partial L}{\partial z}\right] \tag{8}$$

Thus, the final gradient estimation becomes:

$$\nabla_\theta \mathbb{E}_\epsilon[L(z)] = \mathbb{E}_\epsilon\left[\frac{\partial L}{\partial z} \cdot \nabla_\theta \mu(\theta)\right]$$
$$= \mathbb{E}_\epsilon\left[L(z) \cdot \frac{\epsilon}{\sigma} \cdot \nabla_\theta \mu(\theta)\right] \tag{9}$$

### A.1.2. PROOF USING STEIN'S LEMMA

Let $\mu(\theta) = f(\theta; x)$, so $z = \mu(\theta) + \sigma\epsilon$. We need to compute $\mathbb{E}_\epsilon\left[L(z) \cdot \frac{\epsilon}{\sigma}\right]$.

Stein's lemma states that for a standard normal random variable $\epsilon \sim \mathcal{N}(0, 1)$ and any sufficiently smooth function $h$:

$$\mathbb{E}_\epsilon[\epsilon \cdot h(\epsilon)] = \mathbb{E}_\epsilon\left[\frac{\partial h}{\partial \epsilon}\right] \tag{10}$$

In our case, let $h(\epsilon) = L(\mu(\theta) + \sigma\epsilon)$. Then:

$$\frac{\partial h}{\partial \epsilon} = \frac{\partial L}{\partial z} \cdot \frac{\partial z}{\partial \epsilon} = \frac{\partial L}{\partial z} \cdot \sigma \tag{11}$$

Applying Stein's lemma:

$$\mathbb{E}_\epsilon[\epsilon \cdot L(\mu(\theta) + \sigma\epsilon)] = \mathbb{E}_\epsilon\left[\frac{\partial L}{\partial z} \cdot \sigma\right] \tag{12}$$

Therefore:

$$\mathbb{E}_\epsilon\left[L(z) \cdot \frac{\epsilon}{\sigma}\right] = \mathbb{E}_\epsilon\left[\frac{\partial L}{\partial z}\right] \tag{13}$$

### A.1.3. ALTERNATIVE PROOF VIA INTEGRATION BY PARTS

We can also derive this result through direct integration. First, we express the expectation:

$$\mathbb{E}_\epsilon\left[L(z) \cdot \frac{\epsilon}{\sigma}\right] = \int_{-\infty}^{\infty} L(\mu(\theta) + \sigma\epsilon) \cdot \frac{\epsilon}{\sigma} \cdot \frac{1}{\sqrt{2\pi}} e^{-\frac{\epsilon^2}{2}} d\epsilon \tag{14}$$

Using the substitution $z = \mu(\theta) + \sigma\epsilon$, we have $\epsilon = \frac{z - \mu(\theta)}{\sigma}$ and $d\epsilon = \frac{1}{\sigma} dz$. Substituting:

$$\mathbb{E}_\epsilon\left[L(z) \cdot \frac{\epsilon}{\sigma}\right] = \int_{-\infty}^{\infty} L(z) \cdot \frac{z - \mu(\theta)}{\sigma^2}$$
$$\cdot \frac{1}{\sqrt{2\pi}\sigma} e^{-\frac{(z-\mu(\theta))^2}{2\sigma^2}} dz \tag{15}$$

The term $\frac{z-\mu(\theta)}{\sigma^2} \cdot \frac{1}{\sqrt{2\pi}\sigma} e^{-\frac{(z-\mu(\theta))^2}{2\sigma^2}}$ can be rewritten as:

$$\frac{z - \mu(\theta)}{\sigma^2} \cdot p(z) = -\frac{\partial}{\partial z} p(z) \tag{16}$$

where $p(z) = \frac{1}{\sqrt{2\pi}\sigma} e^{-\frac{(z-\mu(\theta))^2}{2\sigma^2}}$ is the probability density function of $\mathcal{N}(\mu(\theta), \sigma^2)$.

Thus, our expression becomes:

$$\mathbb{E}_\epsilon\left[L(z) \cdot \frac{\epsilon}{\sigma}\right] = -\int_{-\infty}^{\infty} L(z) \cdot \frac{\partial}{\partial z} p(z) \, dz \tag{17}$$

Applying integration by parts with $u = L(z)$ and $dv = \frac{\partial}{\partial z} p(z) \, dz$:

$$-\int_{-\infty}^{\infty} L(z) \cdot \frac{\partial}{\partial z} p(z) \, dz = -[L(z) \cdot p(z)]_{-\infty}^{\infty}$$
$$+ \int_{-\infty}^{\infty} \frac{\partial L}{\partial z} \cdot p(z) \, dz \tag{18}$$

Since $\lim_{z \to \pm\infty} L(z) \cdot p(z) = 0$ under mild conditions on $L$, we have:

$$\mathbb{E}_\epsilon\left[L(z) \cdot \frac{\epsilon}{\sigma}\right] = \int_{-\infty}^{\infty} \frac{\partial L}{\partial z} \cdot p(z) \, dz = \mathbb{E}_\epsilon\left[\frac{\partial L}{\partial z}\right] \tag{19}$$

## A.2. Case II: Gradient-Agnostic Loss Functions

### A.2.1. ASSUMPTIONS

We consider the following setting:

- The output is modeled as a Gaussian random variable: $z \sim \mathcal{N}(\mu(\theta), \sigma^2)$, where $\mu(\theta) := f(\theta; x)$.

- The loss function $L(z, y)$ is gradient-agnostic with respect to $z$, meaning it does not provide a useful or existing gradient signal (e.g., the piecewise-constant 0-1 loss).

- The label $y$ is observed and fixed, and does not depend on the random variable $\epsilon$. Therefore, the loss function $L(z, y)$ can be written as $L(z)$ without loss of generality during expectation over $\epsilon$.

- The gradient $\nabla_\theta \mu(\theta)$ exists and is continuous.

### A.2.2. GRADIENT ESTIMATION VIA LIKELIHOOD-RATIO TRICK

We aim to estimate the gradient of $\mathbb{E}_{z \sim p(z;\theta)}[L(z)]$ with respect to $\theta$, where:

$$z = f(\theta; x) + \sigma\epsilon = \mu(\theta) + \sigma\epsilon, \quad \epsilon \sim \mathcal{N}(0,1). \quad (20)$$

The probability density function (PDF) of $z$ is then:

$$p(z;\theta) = \frac{1}{\sqrt{2\pi\sigma^2}} \exp\left(-\frac{(z - \mu(\theta))^2}{2\sigma^2}\right). \quad (21)$$

Using the likelihood-ratio trick, the gradient of the expectation of the loss is given by:

$$\nabla_\theta \mathbb{E}_{z \sim p(z;\theta)}[L(z)] = \mathbb{E}_{z \sim p(z;\theta)}[L(z) \cdot \nabla_\theta \log p(z;\theta)]. \quad (22)$$

Now, compute the derivative of the log of the Gaussian PDF with respect to $\theta$:

$$\nabla_\theta \log p(z;\theta) = \nabla_\theta\left[-\frac{1}{2}\log(2\pi\sigma^2) - \frac{(z - \mu(\theta))^2}{2\sigma^2}\right] \quad (23)$$

$$= \nabla_\theta\left[-\frac{(z - \mu(\theta))^2}{2\sigma^2}\right] \quad (24)$$

$$= \frac{(z - \mu(\theta))}{\sigma^2} \cdot \nabla_\theta \mu(\theta) \quad (25)$$

Substitute this back into the gradient formula:

$$\nabla_\theta \mathbb{E}_{z \sim p(z;\theta)}[L(z)]$$
$$= \mathbb{E}_{z \sim p(z;\theta)}\left[L(z) \cdot \frac{z - \mu(\theta)}{\sigma^2} \cdot \nabla_\theta \mu(\theta)\right] \quad (26)$$

Since $z = \mu(\theta) + \sigma\epsilon$, we can rewrite the expectation in terms of $\epsilon$:

$$\nabla_\theta \mathbb{E}_{z \sim p(z;\theta)}[L(z)]$$
$$= \mathbb{E}_{\epsilon \sim \mathcal{N}(0,1)}\left[L(\mu(\theta) + \sigma\epsilon) \cdot \frac{\epsilon}{\sigma} \cdot \nabla_\theta \mu(\theta)\right] \quad (27)$$

Thus, the gradient of the expectation of the loss with respect to $\theta$ is:

$$\nabla_\theta \mathbb{E}_{z \sim p(z;\theta)}[L(z)] = \mathbb{E}_{\epsilon \sim \mathcal{N}(0,1)}\left[L(z) \cdot \frac{\epsilon}{\sigma} \cdot \nabla_\theta \mu(\theta)\right]. \quad (28)$$

This expression provides an unbiased estimator of the gradient, even when $L$ is gradient-agnostic (i.e., when $\frac{\partial L}{\partial z}$ is zero, undefined, or otherwise uninformative).

### A.3. Summary

• If $L(z)$ provides an informative gradient (is differentiable), the reparameterization trick gives:

$$z = \mu(\theta) + \sigma\epsilon, \quad \epsilon \sim \mathcal{N}(0,1) \quad (29)$$

$$\nabla_\theta \mathbb{E}_{z \sim p(z;\theta)}[L(z)]$$
$$= \mathbb{E}_\epsilon\left[\frac{\partial L}{\partial z} \cdot \nabla_\theta \mu(\theta)\right] \quad (30)$$
$$= \mathbb{E}_\epsilon\left[L(z) \cdot \frac{\epsilon}{\sigma} \cdot \nabla_\theta \mu(\theta)\right]$$

• If $L(z)$ is gradient-agnostic (e.g., piecewise-constant), the likelihood-ratio trick provides a valid and unbiased estimator:

$$\nabla_\theta \mathbb{E}_{z \sim p(z;\theta)}[L(z)]$$
$$= \mathbb{E}_{\epsilon \sim \mathcal{N}(0,1)}\left[L(z) \cdot \frac{\epsilon}{\sigma} \cdot \nabla_\theta \mu(\theta)\right] \quad (31)$$

**Key Insight**: The OVLR estimator takes the same form in both cases, demonstrating its unified nature. It does not require the loss function $L$ to be differentiable, only that the mapping $\mu(\theta)$ is. This allows OVLR to optimize gradient-agnostic objectives directly while maintaining computational efficiency through the use of vector-Jacobian products on $\nabla_\theta \mu(\theta)$.

## B. Variance, Stability, and Convergence Analysis for the OVLR Estimator

### B.1. Gradient Estimator Review

We consider the stochastic model

$$z = \mu(\theta) + \sigma\epsilon, \quad \epsilon \sim \mathcal{N}(0,1), \quad (32)$$

with a loss function $L(z)$. The objective is

$$F(\theta) := \mathbb{E}_{z \sim p(z;\theta)}[L(z)], \quad (33)$$

and we use the likelihood-ratio trick to estimate its gradient:

$$\nabla_\theta F(\theta) = \mathbb{E}_{\epsilon \sim \mathcal{N}(0,1)}\left[L(z) \cdot \frac{\epsilon}{\sigma} \cdot \nabla_\theta \mu(\theta)\right]. \quad (34)$$

The corresponding OVLR Monte Carlo estimator with $n$ samples is

$$g(\epsilon) := L(z) \cdot \frac{\epsilon}{\sigma} \cdot \nabla_\theta \mu(\theta),$$
$$\hat{g}_n := \frac{1}{n}\sum_{i=1}^{n} g(\epsilon_i), \quad \epsilon_i \overset{\text{i.i.d.}}{\sim} \mathcal{N}(0,1). \quad (35)$$

## B.2. Variance Analysis

**Assumption B.1** (Bounded loss)**.** There exists a constant $M > 0$ such that $|L(z)| \leq M$ almost surely.

**Assumption B.2** (Bounded model gradient)**.** There exists a constant $G > 0$ such that $\|\nabla_\theta \mu(\theta)\| \leq G$.

*Remark* B.3. The bounded-loss assumption is essential for our variance and convergence guarantees. In settings with unbounded losses (e.g., certain reinforcement learning problems with unbounded reward functions), additional conditions such as sub-Gaussian tails or moment bounds would be required. Extending the theoretical framework to such regimes is an important direction for future work but lies beyond the scope of this paper.

By the linearity of expectation, $\hat{g}_n$ is unbiased:

$$\mathbb{E}[\hat{g}_n] = \frac{1}{n} \sum_{i=1}^{n} \mathbb{E}[g(\epsilon_i)] = \mathbb{E}[g(\epsilon)] = \nabla_\theta F(\theta). \quad (36)$$

**Theorem B.4** (Covariance upper bound)**.** *Under Assumptions B.1 and B.2, the covariance matrix of the estimator satisfies*

$$\mathrm{Cov}(\hat{g}_n) \preceq \frac{M^2}{n\sigma^2} \nabla_\theta \mu(\theta) \nabla_\theta \mu(\theta)^\top, \quad (37)$$

*and its mean-squared error obeys*

$$\mathbb{E}\|\hat{g}_n - \nabla_\theta F(\theta)\|^2 \leq \frac{M^2 G^2}{n\sigma^2}, \quad (38)$$

*where $\preceq$ denotes the Loewner partial order.*

*Proof.* Observe that

$$g(\epsilon)g(\epsilon)^\top = L(\mu + \sigma\epsilon)^2 \frac{\epsilon^2}{\sigma^2} \nabla_\theta \mu \nabla_\theta \mu^\top. \quad (39)$$

Taking expectations and using Assumption B.1,

$$\mathbb{E}[g(\epsilon)g(\epsilon)^\top] = \frac{1}{\sigma^2} \nabla_\theta \mu \nabla_\theta \mu^\top \, \mathbb{E}[L(\mu + \sigma\epsilon)^2 \epsilon^2]$$
$$\preceq \frac{M^2}{\sigma^2} \nabla_\theta \mu \nabla_\theta \mu^\top, \quad (40)$$

because $\mathbb{E}[\epsilon^2] = 1$ and $L^2 \leq M^2$.

Since $\mathrm{Cov}(g(\epsilon)) = \mathbb{E}[g(\epsilon)g(\epsilon)^\top] - \nabla_\theta F(\theta)\nabla_\theta F(\theta)^\top \preceq \mathbb{E}[g(\epsilon)g(\epsilon)^\top]$, we obtain

$$\mathrm{Cov}(\hat{g}_n) = \frac{1}{n} \mathrm{Cov}(g(\epsilon)) \preceq \frac{M^2}{n\sigma^2} \nabla_\theta \mu \nabla_\theta \mu^\top. \quad (41)$$

For the second claim,

$$\mathbb{E}\|\hat{g}_n - \nabla_\theta F(\theta)\|^2 = \mathrm{tr}(\mathrm{Cov}(\hat{g}_n))$$
$$\leq \mathrm{tr}\left(\frac{M^2}{n\sigma^2} \nabla_\theta \mu \nabla_\theta \mu^\top\right) \quad (42)$$
$$= \frac{M^2}{n\sigma^2} \|\nabla_\theta \mu\|^2 \leq \frac{M^2 G^2}{n\sigma^2},$$

where the last inequality uses Assumption B.2. □

*Remark* B.5. The variance scales as $O(1/(n\sigma^2))$. Increasing the noise scale $\sigma$ reduces the variance, but an excessively large $\sigma$ makes the output too noisy and can degrade the accuracy of the gradient direction.

## B.3. Stability Analysis

We assess stability through the Lipschitz continuity of the estimator $g(\theta, \epsilon)$ with respect to the noise $\epsilon$.

**Assumption B.6** (Lipschitz loss)**.** The loss function $L$ is $K$-Lipschitz continuous, i.e., $|L(x) - L(y)| \leq K|x - y|$ for all $x, y$.

**Theorem B.7** (Lipschitz stability of the estimator)**.** *Under Assumptions B.1 and B.6, for any two noise realizations $\epsilon_1, \epsilon_2$ belonging to a bounded interval $[-c, c]$ (e.g., $c = 3$ covering 99.7% of the standard normal mass), the estimator satisfies*

$$\|g(\theta, \epsilon_1) - g(\theta, \epsilon_2)\| \leq \|\nabla_\theta \mu(\theta)\| \left(Kc + \frac{M}{\sigma}\right) |\epsilon_1 - \epsilon_2|. \quad (43)$$

*Thus, on the interval $[-c, c]$, $g(\theta, \cdot)$ is Lipschitz continuous with constant $L_g = G(Kc + M/\sigma)$.*

*Proof.* Write $z_i = \mu(\theta) + \sigma\epsilon_i$ for $i = 1, 2$. Then

$$\|g(\theta, \epsilon_1) - g(\theta, \epsilon_2)\|$$
$$= \left\|L(z_1)\frac{\epsilon_1}{\sigma}\nabla_\theta \mu(\theta) - L(z_2)\frac{\epsilon_2}{\sigma}\nabla_\theta \mu(\theta)\right\| \quad (44)$$
$$= \|\nabla_\theta \mu(\theta)\| \left|L(z_1)\frac{\epsilon_1}{\sigma} - L(z_2)\frac{\epsilon_2}{\sigma}\right|.$$

Decompose the difference as

$$\left|L(z_1)\frac{\epsilon_1}{\sigma} - L(z_2)\frac{\epsilon_2}{\sigma}\right| = \left|\frac{\epsilon_1}{\sigma}\big(L(z_1) - L(z_2)\big)\right.$$
$$\left. + L(z_2)\frac{\epsilon_1 - \epsilon_2}{\sigma}\right| \quad (45)$$
$$\leq \frac{|\epsilon_1|}{\sigma}|L(z_1) - L(z_2)|$$
$$+ |L(z_2)|\frac{|\epsilon_1 - \epsilon_2|}{\sigma}.$$

Using the Lipschitz property and boundedness of $L$,

$$\frac{|\epsilon_1|}{\sigma}|L(z_1) - L(z_2)| \leq \frac{|\epsilon_1|}{\sigma} K\sigma|\epsilon_1 - \epsilon_2| = K|\epsilon_1| |\epsilon_1 - \epsilon_2|, \quad (46)$$

and

$$|L(z_2)|\frac{|\epsilon_1 - \epsilon_2|}{\sigma} \leq \frac{M}{\sigma}|\epsilon_1 - \epsilon_2|. \quad (47)$$

Combining the bounds and noting that $|\epsilon_1| \leq c$ on the considered interval gives

$$\|g(\theta, \epsilon_1) - g(\theta, \epsilon_2)\| \leq \|\nabla_\theta \mu(\theta)\| \left(Kc + \frac{M}{\sigma}\right)|\epsilon_1 - \epsilon_2|. \quad (48)$$

The second statement follows directly from Assumption B.2. □

## B.4. Convergence Analysis

**Assumption B.8** (Smoothness and strong convexity)**.** The objective $F$ is $L$-smooth and $\mu$-strongly convex, i.e.,

$$\|\nabla_\theta F(\theta) - \nabla_\theta F(\theta')\| \leq L\|\theta - \theta'\|, \qquad (49)$$

$$F(\theta') \geq F(\theta) + \langle \nabla_\theta F(\theta), \theta' - \theta \rangle + \frac{\mu}{2}\|\theta' - \theta\|^2, \quad (50)$$

for all $\theta, \theta'$.

**Assumption B.9** (Learning-rate schedule)**.** The step sizes follow

$$\eta_k = \eta_0\, k^{-(1/2+\epsilon)}, \qquad k \geq 1, \qquad (51)$$

with constants $\eta_0 > 0$ and $\epsilon \in (0, 1/2)$. This choice satisfies

$$\sum_{k=1}^{\infty} \eta_k = \infty, \qquad \sum_{k=1}^{\infty} \eta_k^2 < \infty. \qquad (52)$$

**Lemma B.10** (Gradient lower bound under strong convexity)**.** *Under Assumption B.8,*

$$\|\nabla_\theta F(\theta)\|^2 \geq 2\mu\big(F(\theta) - F^*\big), \qquad F^* := \min_\theta F(\theta). \qquad (53)$$

*Proof.* Let $\theta^* = \arg\min_\theta F(\theta)$. Strong convexity gives

$$\langle \nabla_\theta F(\theta), \theta - \theta^* \rangle \geq F(\theta) - F^* + \frac{\mu}{2}\|\theta - \theta^*\|^2. \quad (54)$$

By the Cauchy–Schwarz inequality,

$$\|\nabla_\theta F(\theta)\|\,\|\theta - \theta^*\| \geq F(\theta) - F^* + \frac{\mu}{2}\|\theta - \theta^*\|^2. \quad (55)$$

Set $g = \|\nabla_\theta F(\theta)\|$, $s = \|\theta - \theta^*\|$, $\delta = F(\theta) - F^*$. The inequality becomes $gs \geq \delta + \frac{\mu}{2}s^2$. Using $(g - \mu s)^2 \geq 0$, we have $g^2 \geq 2\mu g s - \mu^2 s^2 \geq 2\mu(\delta + \frac{\mu}{2}s^2) - \mu^2 s^2 = 2\mu\delta$. Hence $\|\nabla_\theta F(\theta)\|^2 \geq 2\mu\delta$. $\square$

**Theorem B.11** (Convergence rate of OVLR)**.** *Under Assumptions B.1–B.9, consider the update*

$$\begin{aligned}
\theta_{k+1} &= \theta_k - \eta_k \hat{g}_{n,k}, \\
\eta_k &= \eta_0 k^{-(1/2+\epsilon)}, \qquad (56) \\
0 &< \epsilon < \tfrac{1}{2}, \quad 0 < \eta_0 < \tfrac{1}{L}.
\end{aligned}$$

*where*

$$\hat{g}_{n,k} = \frac{1}{n}\sum_{i=1}^{n} L\big(\mu(\theta_k) + \sigma\epsilon_{i,k}\big)\frac{\epsilon_{i,k}}{\sigma}\nabla_\theta\mu(\theta_k), \qquad (57)$$

$$\epsilon_{i,k} \overset{i.i.d.}{\sim} \mathcal{N}(0,1).$$

*Define the weighted average*

$$\bar{\theta}_K := \frac{\sum_{k=0}^{K-1}\eta_k\theta_k}{\sum_{k=0}^{K-1}\eta_k}. \qquad (58)$$

*Then*

$$\mathbb{E}\big[F(\bar{\theta}_K)\big] - F^* \leq \frac{C}{K^{1/2-\epsilon}}, \qquad (59)$$

*where*

$$C = \frac{\dfrac{G_0}{\mu} + \dfrac{L\sigma_g^2}{2\mu}\eta_0^2\big(1 + \dfrac{1}{2\epsilon}\big)}{\eta_0\big(\tfrac{1}{2} - \epsilon\big)}, \qquad (60)$$

$$G_0 := \mathbb{E}[F(\theta_0)] - F^*, \quad \sigma_g^2 := \frac{M^2 G^2}{n\sigma^2}.$$

*Proof.* Using the $L$-smoothness of $F$,

$$F(\theta_{k+1}) \leq F(\theta_k) - \eta_k\langle\nabla_\theta F(\theta_k), \hat{g}_{n,k}\rangle + \frac{L\eta_k^2}{2}\|\hat{g}_{n,k}\|^2. \qquad (61)$$

Taking expectations and employing unbiasedness $\mathbb{E}[\hat{g}_{n,k} \mid \theta_k] = \nabla_\theta F(\theta_k)$,

$$\begin{aligned}
\mathbb{E}[F(\theta_{k+1}) \mid \theta_k] &\leq F(\theta_k) - \eta_k\|\nabla_\theta F(\theta_k)\|^2 \\
&\quad + \frac{L\eta_k^2}{2}\mathbb{E}\big[\|\hat{g}_{n,k}\|^2 \mid \theta_k\big].
\end{aligned} \qquad (62)$$

From Theorem B.4,

$$\mathbb{E}\big[\|\hat{g}_{n,k}\|^2 \mid \theta_k\big] \leq \|\nabla_\theta F(\theta_k)\|^2 + \sigma_g^2. \qquad (63)$$

Thus,

$$\begin{aligned}
\mathbb{E}[F(\theta_{k+1}) \mid \theta_k] &\leq F(\theta_k) - \eta_k\Big(1 - \frac{L\eta_k}{2}\Big)\|\nabla_\theta F(\theta_k)\|^2 \\
&\quad + \frac{L\eta_k^2}{2}\sigma_g^2.
\end{aligned} \qquad (64)$$

Because $\eta_k \leq \eta_0 < 1/L$, we have $1 - L\eta_k/2 \geq 1/2$, and therefore

$$\mathbb{E}[F(\theta_{k+1}) \mid \theta_k] \leq F(\theta_k) - \frac{\eta_k}{2}\|\nabla_\theta F(\theta_k)\|^2 + \frac{L\eta_k^2}{2}\sigma_g^2. \qquad (65)$$

Taking full expectation and applying Lemma B.10,

$$\begin{aligned}
G_{k+1} &\leq G_k - \mu\eta_k G_k + \frac{L\eta_k^2}{2}\sigma_g^2, \\
G_k &:= \mathbb{E}[F(\theta_k)] - F^*.
\end{aligned} \qquad (66)$$

Rearranging gives

$$G_{k+1} - G_k \leq -\mu\eta_k G_k + \frac{L\eta_k^2}{2}\sigma_g^2. \qquad (67)$$

Summing from $k = 0$ to $K-1$ and using $G_K \geq 0$,

$$\sum_{k=0}^{K-1}\eta_k G_k \leq \frac{G_0}{\mu} + \frac{L\sigma_g^2}{2\mu}\sum_{k=0}^{K-1}\eta_k^2. \qquad (68)$$

For the chosen step-size schedule,

$$S_K := \sum_{k=0}^{K-1} \eta_k \geq \frac{\eta_0}{\frac{1}{2} - \epsilon} \left( K^{\frac{1}{2} - \epsilon} - 1 \right),$$

$$Q_K := \sum_{k=0}^{K-1} \eta_k^2 \leq \eta_0^2 \left( 1 + \frac{1}{2\epsilon} \right). \tag{69}$$

Consequently,

$$\mathbb{E}\left[ F(\bar{\theta}_K) \right] - F^* = \frac{\sum_{k=0}^{K-1} \eta_k G_k}{S_K}$$

$$\leq \frac{\frac{G_0}{\mu} + \frac{L\sigma_g^2}{2\mu} \eta_0^2 \left( 1 + \frac{1}{2\epsilon} \right)}{\frac{\eta_0}{\frac{1}{2} - \epsilon} \left( K^{\frac{1}{2} - \epsilon} - 1 \right)} \tag{70}$$

$$\leq \frac{C}{K^{\frac{1}{2} - \epsilon}},$$

with the constant $C$ defined in the theorem statement. $\square$

**Corollary B.12.** *As $\epsilon \to 0^+$, the convergence rate approaches $O(1/\sqrt{K})$.*

### B.5. Smoothness of the Expected Objective for Non-smooth Losses

A potential concern is whether Assumption B.8(Smoothness) holds for gradient-agnostic objectives like the $0 - 1$ loss. We clarify that OVLR optimizes the expected loss $F(\theta) = \mathbb{E}_{z \sim \mathcal{N}(\mu(\theta), \sigma^2)}[L(z)]$, which can be expressed as a convolution:

$$F(\theta) = (L * G_\sigma)(\mu(\theta))$$

$$= \int_{\mathbb{R}^d} L(u) \frac{1}{(2\pi\sigma^2)^{d/2}} \exp\left( -\frac{\|u - \mu(\theta)\|^2}{2\sigma^2} \right) du \tag{71}$$

where $G_\sigma$ is the Gaussian kernel. Since $G_\sigma$ is infinitely differentiable ($\mathcal{C}^\infty$), the convolution $F$ inherits this smoothness regardless of the continuity of $L$, provided $L$ is bounded. Thus, the gradient $\nabla_\theta F(\theta)$ is well-defined and Lipschitz continuous, satisfying the prerequisite for standard convergence analysis.

### B.6. Convergence under Relaxed Non-convex Assumptions

To address the non-convex nature of deep neural networks, we provide a convergence guarantee to first-order stationary points.

**Corollary B.13.** *Under Assumptions B.1 B.2, and B.9, if $F(\theta)$ is L-smooth and bounded from below by $F^*$, a constant step size $\eta < 1/L$ yields:*

$$\min_{k=0,\ldots,K-1} \mathbb{E}[\|\nabla F(\theta_k)\|^2] \leq \frac{2(F(\theta_0) - F^*)}{\eta K} + \frac{L\eta\sigma_g^2}{2} \tag{72}$$

where $\sigma_g^2 = \frac{M^2 G^2}{n\sigma^2}$ *is the variance bound from Theorem B.4.*

**Proof Sketch:** From the $L$-smoothness of $F$:

$$\mathbb{E}[F(\theta_{k+1})] \leq F(\theta_k) - \eta\|\nabla F(\theta_k)\|^2 + \frac{L\eta^2}{2}(\|\nabla F(\theta_k)\|^2 + \sigma_g^2) \tag{73}$$

Rearranging and summing from $k = 0$ to $K - 1$, we obtain the bound. This ensures that OVLR reliably converges to a stationary point even in complex, non-convex landscapes like those of Transformers and Mamba.

## C. Theoretical Analysis of Robustness in OVLR

This section provides a formal analysis of why OVLR exhibits superior robustness compared to standard backpropagation (BP) when optimizing gradient-agnostic objectives. The key lies in how OVLR transforms a loss function with vanishing or pathological gradients into a smoothed field that retains informative gradient signals.

### C.1. Bounded Gradient Influence in Classification

Standard models trained with Cross-Entropy (CE) loss are prone to noise memorization. The CE gradient, $\nabla_\mathbf{z} L_{CE} = \text{softmax}(\mathbf{z}) - \mathbf{y}$, can grow arbitrarily large for mislabeled samples, forcing the model to overfit corrupted labels.

In contrast, OVLR optimizes the gradient-agnostic Hard 0-1 Loss $L_{0\text{-}1}(\mathbf{z}, y) = \mathbb{I}(\arg\max_k z_k \neq y)$ via its Gaussian-smoothed expectation:

$$\tilde{L}_{0\text{-}1}(\mathbf{z}, y) = \mathbb{E}_{\boldsymbol{\epsilon} \sim \mathcal{N}(\mathbf{0}, \sigma^2 \mathbf{I})}[L_{0\text{-}1}(\mathbf{z} + \boldsymbol{\epsilon}, y)]. \tag{74}$$

The resulting OVLR gradient estimator is

$$\nabla_\theta \tilde{L}_{0\text{-}1} = \mathbb{E}_{\boldsymbol{\epsilon}}\left[ L_{0\text{-}1}(f(\mathbf{x};\theta) + \boldsymbol{\epsilon}, y) \cdot \nabla_\theta \log p(\boldsymbol{\epsilon}) \right]. \tag{75}$$

Crucially, because $|L_{0\text{-}1}| \leq 1$, the gradient magnitude is intrinsically bounded by the score function $\nabla_\theta \log p(\boldsymbol{\epsilon})$, not by the unbounded prediction error. This boundedness explains why OVLR maintains stable test accuracy (73.0%) under 60% label noise, whereas CE overfits and degrades.

### C.2. Expanded Gradient Capture in Regression

For robust regression, the Truncated $L_1$ Loss $L_{\text{Trunc}}(\Delta) = \min(|\Delta|, \tau)$ has a "dead zone" where the analytic gradient vanishes for $|\Delta| > \tau$:

$$\frac{\partial L_{\text{Trunc}}}{\partial \hat{y}} = 0, \quad \forall |\Delta| > \tau. \tag{76}$$

Standard BP with this loss fails to recover the signal (MSE: 4.09361) if the initial prediction falls into this zero-gradient region.

OVLR overcomes this by optimizing the convolved loss $\tilde{L}_{\text{Trunc}} = L_{\text{Trunc}} * G_\sigma$, where $G_\sigma$ is a Gaussian kernel. The gradient becomes

$$\nabla_{\hat{y}} \tilde{L}_{\text{Trunc}}(\Delta) = \int \nabla_{\hat{y}} L_{\text{Trunc}}(u) \, G_\sigma(\Delta - u) \, du. \quad (77)$$

Even when $|\Delta| > \tau$, the convolution with a Gaussian of width $\sigma$ ensures a non-zero gradient as long as the support of $G_\sigma$ overlaps with the region $|u| < \tau$. This effectively expands the capture range from $\tau$ to approximately $\tau + 3\sigma$, enabling OVLR to achieve near-optimal recovery (MSE: 0.00032).

### C.3. Summary

OVLR's robustness stems from two coupled mechanisms: (1) transforming gradient-agnostic losses into smoothed objectives that retain bounded, non-zero gradients, and (2) using output-level perturbations to estimate these gradients efficiently. This combination allows OVLR to optimize objectives where standard BP fails, without sacrificing scalability or stability.

## D. Variance Reduction for Gradient-Agnostic Losses

We evaluated the effectiveness of variance-reduction strategies under the gradient-agnostic 0-1 loss across different repetition factors ($K$). As seen in Figure 7, gradient variance in the vanilla setting (Fig. 7a) grows nearly linearly with $1/\sigma^2$, but the absolute scale is higher compared to the smooth-loss scenario, indicating greater sensitivity to the perturbation noise. Antithetic sampling (Fig. 7b) reduces variance by approximately two orders of magnitude, with a smaller slope and an $R^2$ of 0.59, suggesting partial noise cancellation. Output-level repetition (Fig. 7c) suppresses variance further by another two orders, while increasing the repetition factor to 1000 (Fig. 7g) leads to an additional one-order reduction in variance. Combining repetition and antithetic sampling (Fig. 7d, 7h) yields near-constant variance, almost independent of $1/\sigma^2$, with $R^2$ values approaching 0, indicating that dominant noise sources are effectively neutralized. These results confirm that variance-reduction techniques are even more crucial for gradient-agnostic losses than for smooth ones, with the combination of repetition and antithetic sampling proving most effective in stabilizing gradient estimates.

## E. Effectiveness of OVLR in Training Generative Models

We evaluate the proposed OVLR framework across two fundamental generative modeling paradigms: Generative Adversarial Networks (GANs) and Variational Autoencoders

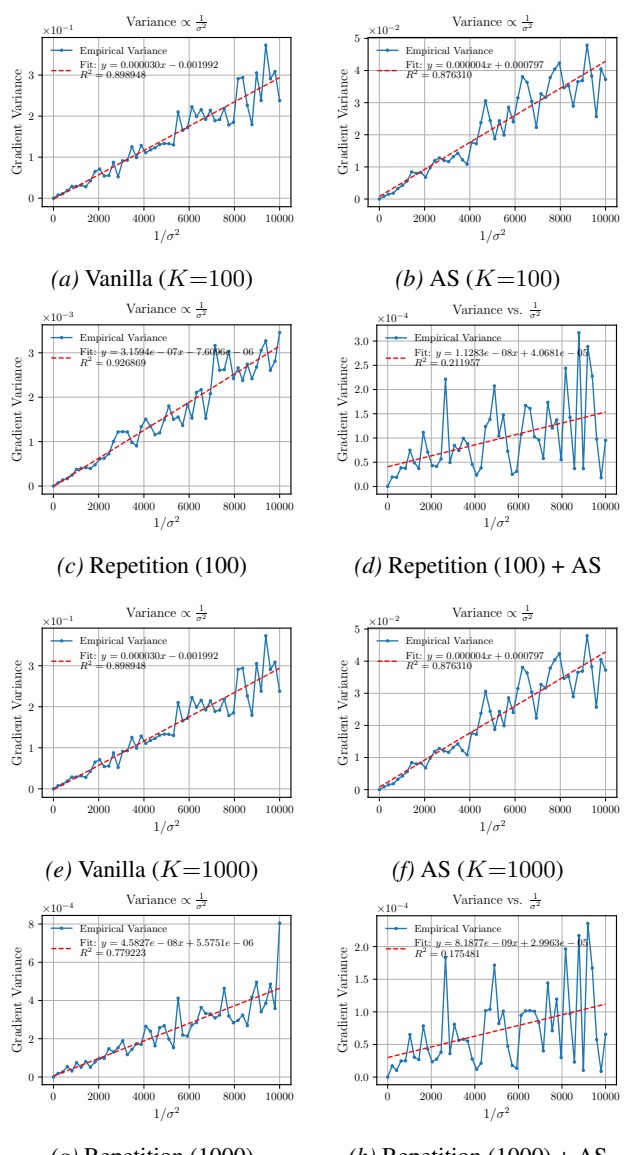

*(a)* Vanilla ($K$=100)  *(b)* AS ($K$=100)

*(c)* Repetition (100)  *(d)* Repetition (100) + AS

*(e)* Vanilla ($K$=1000)  *(f)* AS ($K$=1000)

*(g)* Repetition (1000)  *(h)* Repetition (1000) + AS

*Figure 7.* Gradient Variance under the Gradient-Agnostic 0–1 Loss across All Sampling Strategies and Repetition Factors ($K$).

(VAEs). Our goal is to verify if OVLR can successfully estimate gradients for complex, high-dimensional probability distributions and match the performance of standard backpropagation (BP).

### E.1. Generative Adversarial Network (GAN)

We conducted experiments on the MNIST dataset comparing OVLR with the conventional BP approach under a standard GAN (Goodfellow et al., 2014) training procedure. Both methods utilized the same architecture, optimized via Adam ($lr = 10^{-4}$, $\beta_1 = 0.5$, $\beta_2 = 0.999$) with a batch size of 128 and a latent dimension of 100. Training spanned 200 epochs with identical random seeds to ensure a fair comparison. For OVLR, we employed a Gaussian noise distribution with antithetic sampling to estimate the generator gradients.

**Qualitative Analysis.** To visually assess sample quality, Figure 8 presents randomly generated images at epoch 200, batch 500. At this stage, both methods produce visually similar digits with comparable diversity. No significant qualitative degradation is observed in the OVLR-trained model, suggesting it successfully captures the target distribution.

**Quantitative Analysis.** As summarized in Table 10, we use the Fréchet Inception Distance (FID) (Heusel et al., 2017) to provide a rigorous evaluation under an aligned protocol. For the GAN architecture, OVLR achieves an FID of 40.27, outperforming BP's 53.33 by 13.06 points. This advantage is consistent with established findings that output-space smoothing stabilizes adversarial training. For the VAE, OVLR and BP are effectively tied (29.80 vs. 29.73), as the single-objective VAE setting does not benefit from the regularization that helps GANs.

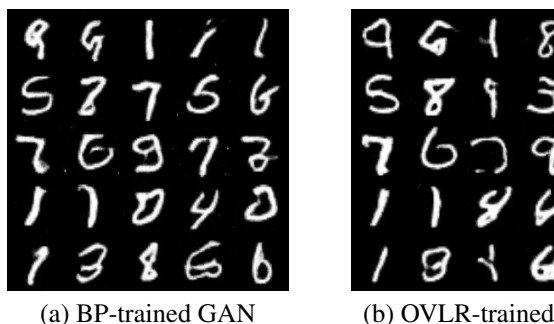

(a) BP-trained GAN  (b) OVLR-trained GAN

*Figure 8.* Comparison of Generated Samples at Epoch 200, Batch 0 Using BP (left) and OVLR (right) on the MNIST Dataset.

### E.2. Variational Autoencoder (VAE)

We further validate OVLR using a standard VAE (Kingma & Welling, 2014) testbed. The architecture employs a two-layer encoder-decoder with a 20-dimensional latent space and a hidden dimension of 400. Both methods were trained for 200 epochs with a batch size of 128 and a learning

*Table 10.* Quantitative comparison of generative models on MNIST. Lower FID indicates better sample quality.

| Method | Model | FID ↓ |
|--------|-------|-------|
| BP | GAN | 53.33 |
| OVLR | GAN | 40.27 |
| BP | VAE | 29.73 |
| OVLR | VAE | 29.80 |

rate of $10^{-3}$. For OVLR, we employed a Gaussian noise distribution with antithetic sampling.

**Reconstruction and Generation.** Figure 9 illustrates the reconstruction performance. Both training schemes successfully recover the primary structures of input images, although both exhibit the characteristic slight blurring of VAEs. Furthermore, Figure 10 demonstrates samples drawn from the prior, showing that OVLR effectively learns a meaningful and diverse latent space.

**Quantitative Comparison.** Quantitative results in Table 10 show that OVLR achieves an FID of 29.80 compared to BP's 29.73 on VAE. The two methods are effectively tied, confirming that OVLR is a viable alternative for optimizing latent variable models without relying on traditional backpropagation through the sampling process.

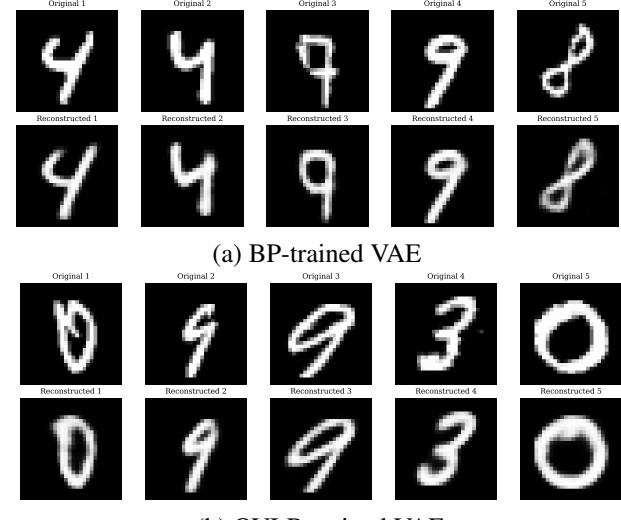

(a) BP-trained VAE

(b) OVLR-trained VAE

*Figure 9.* Reconstruction Results. Top: Input image; Bottom: Reconstructed Image.

### E.3. Discussion on Generative Modeling Results

The generative modeling results demonstrate that OVLR provides sufficiently informative gradients to train complex, high-dimensional models like GANs and VAEs. Under the aligned evaluation protocol, OVLR achieves a lower FID than BP on GANs (40.27 vs. 53.33), consistent with

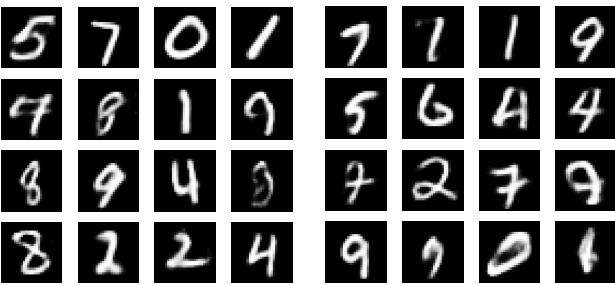

    (a) BP-trained VAE      (b) OVLR-trained VAE

*Figure 10.* Samples Generated from Latent Space.

established findings that output-space smoothing stabilizes adversarial training (Goodfellow et al., 2014). On VAEs, OVLR and BP are effectively tied (29.80 vs. 29.73), as the single-objective VAE setting does not benefit from the regularization that helps GANs.

These results confirm that OVLR is compatible with standard generative training, and can even improve over BP in adversarial settings. Its primary advantage, however, lies in enabling optimization where BP is inapplicable—as demonstrated in Section 5.3 and Section 5.4 on robust, gradient-agnostic losses.

## F. Effectiveness of OVLR in Training Language Models

### F.1. Selective Copying

The Selective Copying task is a synthetic benchmark for evaluating sequence modeling capabilities, particularly designed to assess the memorization ability of recurrent models (Gu & Dao, 2024). In this task, the model is required to store and reproduce a sequence of tokens after a long delay period, posing significant challenges for long-context modeling. In our experimental setting, we consider sequences of length 4096, with a vocabulary size of 16 tokens, including a designated "noise" token. The model is tasked to memorize and reproduce 16 specific "data" tokens embedded within the sequence. All methods are configured with 2 layers and a mamba (Gu & Dao, 2024) model dimension of $D = 64$. Training is conducted for 300K steps using a constant learning rate of 0.0001 and a batch size of 32. For the OVLR estimator, we adopt a symmetric noise distribution with a noise scale of $\sigma = 1.0$, and set the number of repeated samples to 200.

To investigate the effectiveness of our proposed OVLR in training recurrent models, we apply both standard backpropagation (BP) and OVLR to train the Mamba v1 architecture on this task. Our objective is to evaluate whether OVLR can achieve comparable learning outcomes to traditional BP methods in such challenging settings. As shown in Table 11,

both BP and OVLR successfully solve the Copying task, achieving near-perfect performance on the validation set. This result empirically confirms that OVLR does not hinder the model's ability to learn long-term dependencies, and performs on par with standard BP in this synthetic benchmark.

*Table 11.* Validation Accuracy (%) on the Copying task (Sequence Length 4096, Vocab Size 16). Both methods achieve nearly 100% accuracy, demonstrating the effectiveness of OVLR in training recurrent models.

| Method | Validation Accuracy (%) |
| --- | --- |
| BP | 99.8 |
| OVLR | 99.8 |

### F.2. Sentiment Classification

We evaluate our proposed OVLR training method on the IMDB sentiment classification task (Maas et al., 2011) using the BERT-base model (Devlin et al., 2019) as backbone. We compare the standard backpropagation (BP) training with our OVLR-based optimization. Both models are trained for 3 epochs using AdamW optimizer with a learning rate of $2 \times 10^{-5}$ and a batch size of 16. The maximum sequence length is set to 256. All experiments are conducted on a single NVIDIA GPU. To ensure fair comparison, we use the same initialization and random seed across methods. For the OVLR estimator, we adopt a symmetric noise distribution with a noise scale of $\sigma = 1.0$, and set the number of repeated samples to 200. Table 12 summarizes the experimental results on IMDB sentiment classification. The model trained with OVLR achieves comparable performance to BP, with only a slight drop in validation accuracy and similar computational costs. This demonstrates that OVLR can be applied to this task as an alternative optimization strategy without significant performance loss.

*Table 12.* Performance Comparison between BP and OVLR on IMDB Sentiment Classification.

| Method | Validation Accuracy (%) | Training Time per Epoch (s) | Evaluation Time (s) |
| --- | --- | --- | --- |
| BP | 92.43 | 244.0 | 128.0 |
| OVLR | 91.91 | 251.0 | 128.0 |

## G. Effectiveness of OVLR for Robot Manipulation Training

We evaluate the OVLR framework on bimanual robot manipulation using the Aloha simulation suite (Zhao et al., 2023). To benchmark OVLR as a general-purpose gradient

estimator, we adopt the standard L1 regression loss on robot joint positions, which is the common practice in ACT (Zhao et al., 2023) and other imitation learning methods. We focus on two high-precision tasks: Cube Transfer and Bimanual Insertion. Our implementation adopts the Action Chunking Transformer (ACT) backbone, where we replace the standard backpropagation (BP) gradient with the OVLR estimator. We use SGD optimizer with a learning rate of $1 \times 10^{-5}$, momentum of 0.9, a chunking size of 100, and a batch size of 8. Both methods are trained under identical conditions with scripted and human demonstration datasets. Table 13 summarizes the success rates.

*Table 13.* Success rate (%) on Aloha tasks (S: Scripted, H: Human, A: Average). Results indicate that OVLR achieves comparable performance to standard backpropagation in high-precision manipulation tasks.

| Method | Cube Transfer | | | Bimanual Insertion | | |
|---|---|---|---|---|---|---|
| | S | H | A | S | H | A |
| ACT (BP) | 86.0 | 50.0 | 68.0 | 32.0 | 20.0 | 26.0 |
| Ours (OVLR) | 88.0 | 52.0 | 70.0 | 34.0 | 18.0 | 26.0 |

The OVLR-trained policy achieves performance comparable to the BP baseline (overall average success rate of 48.0% versus 47.0%). This demonstrates the key strength of OVLR as a viable gradient estimation alternative: it successfully optimizes high-dimensional policies without relying on gradient signals from a differentiable loss function. While in this imitation learning setting the training signal is dense, our results pave the way for applying OVLR to more challenging robotic learning scenarios where informative gradients are vanishing or inaccessible. This includes direct policy optimization using non-differentiable physics simulators, sparse (e.g., success/failure) rewards, or other forms of black-box feedback that are common in real-world robotic tasks, precisely where standard BP fails to provide a reliable learning signal. The compatibility of OVLR with complex architectures like ACT underscores its practical utility for embodied AI.

## H. Output Dimension Scaling Study

To systematically evaluate OVLR's scalability with output dimensionality, we conduct a controlled study using a single linear layer with $C$ ranging from 10 to 5000, varying the repetition budget from 10 to 2000. Table 14 reports OVLR test accuracy, and Table 15 reports the BP–OVLR accuracy gap.

The results confirm that for $C \leq 2000$, OVLR nearly matches BP at repeat= 2000 (gap $\leq$ 1.0%). Even at $C$=5000, the gap shrinks to 1.81%, indicating that OVLR's cost scales with output dimension rather than parameter count.

*Table 14.* OVLR test accuracy across output dimension $C$ and repeat budget.

| $C$ | rep=10 | rep=50 | rep=200 | rep=500 | rep=1000 | rep=2000 |
|---|---|---|---|---|---|---|
| 10 | 0.750 | 0.767 | 0.775 | 0.783 | 0.775 | 0.767 |
| 20 | 0.733 | 0.733 | 0.729 | 0.725 | 0.721 | 0.725 |
| 50 | 0.617 | 0.653 | 0.667 | 0.660 | 0.667 | 0.667 |
| 100 | 0.498 | 0.559 | 0.596 | 0.600 | 0.593 | 0.598 |
| 200 | 0.363 | 0.470 | 0.508 | 0.506 | 0.518 | 0.512 |
| 500 | 0.213 | 0.375 | 0.429 | 0.444 | 0.449 | 0.454 |
| 1000 | 0.098 | 0.265 | 0.353 | 0.381 | 0.386 | 0.392 |
| 2000 | 0.040 | 0.169 | 0.285 | 0.323 | 0.336 | 0.342 |
| 5000 | 0.007 | 0.056 | 0.165 | 0.227 | 0.255 | 0.273 |

*Table 15.* BP–OVLR accuracy gap (positive = BP better).

| $C$ | rep=10 | rep=50 | rep=200 | rep=500 | rep=1000 | rep=2000 |
|---|---|---|---|---|---|---|
| 10 | 0.017 | 0.000 | -0.008 | -0.017 | -0.008 | 0.000 |
| 50 | 0.050 | 0.013 | 0.000 | 0.007 | 0.000 | 0.000 |
| 100 | 0.100 | 0.039 | 0.003 | -0.002 | 0.006 | 0.001 |
| 500 | 0.243 | 0.081 | 0.026 | 0.012 | 0.007 | 0.002 |
| 1000 | 0.298 | 0.131 | 0.042 | 0.014 | 0.009 | 0.003 |
| 2000 | 0.312 | 0.183 | 0.067 | 0.028 | 0.015 | 0.010 |
| 5000 | 0.284 | 0.235 | 0.126 | 0.064 | 0.036 | 0.018 |

## I. Effect of Noise Distribution Type

We evaluate OVLR with four noise distribution and matching gradient estimator types. All experiments use the same output-level perturbation framework, differing only in the noise distribution and corresponding estimator.

**Gaussian (score-function).** We perturb the deterministic output by adding isotropic Gaussian noise: $\epsilon \sim \mathcal{N}(0, I)$, yielding the stochastic output $\tilde{z} = z + \sigma\epsilon$. The gradient estimator follows the standard score-function form:

$$\hat{g}_{\text{Gauss}} = \frac{1}{B} \sum_{i=1}^{B} L(z + \sigma\epsilon_i, y) \cdot \frac{\epsilon_i}{\sigma}. \tag{78}$$

**Student-t (score-function).** We draw $t \sim \text{StudentT}(df)$ and normalize to unit variance via $\epsilon = t/\sqrt{df/(df-2)}$, then perturb the output as $\tilde{z} = z + \sigma\epsilon$. The negative score function of the normalized noise is:

$$-\nabla_\epsilon \log q(\epsilon) = \frac{(df+1)\,\epsilon}{df-2+\epsilon^2}. \tag{79}$$

The corresponding gradient estimator becomes:

$$\hat{g}_{\text{Student-t}} = \frac{1}{B} \sum_{i=1}^{B} L(z + \sigma\epsilon_i, y) \cdot \frac{-\nabla_\epsilon \log q(\epsilon_i)}{\sigma}. \tag{80}$$

**Laplace (score-function).** We use the zero-mean, unit-variance Laplace distribution $\epsilon \sim \text{Laplace}(0, 1/\sqrt{2})$, so the perturbed output is $\tilde{z} = z + \sigma\epsilon$. The negative score function is piecewise constant:

$$-\nabla_\epsilon \log q(\epsilon) = \sqrt{2}\,\text{sign}(\epsilon). \tag{81}$$

The gradient estimator is then:

$$\hat{g}_{\text{Laplace}} = \frac{1}{B} \sum_{i=1}^{B} L(z + \sigma \epsilon_i, y) \cdot \frac{\sqrt{2} \, \text{sign}(\epsilon_i)}{\sigma}. \quad (82)$$

**Rademacher (two-point SPSA).** Instead of the score-function estimator, we adopt the simultaneous perturbation stochastic approximation (SPSA) with Rademacher directions $\Delta_j \in \{-1, +1\}$ uniformly. The two-point estimator evaluates the loss at both $\pm \sigma \Delta$:

$$\hat{g}_{\text{SPSA}} = \frac{1}{B} \sum_{i=1}^{B} \frac{L(z + \sigma \Delta_i, y) - L(z - \sigma \Delta_i, y)}{2\sigma} \cdot \Delta_i. \quad (83)$$

Unlike the score-function variants above, this estimator uses finite differences rather than the likelihood ratio trick.

*Table 16.* Effect of noise distribution type on OVLR. Accuracy (%) across noise types on MNIST (SimpleCNN) and CIFAR-10 (ResNet-18).

| Dataset | Metric | Gaussian | Student-t | Laplace | Rademacher |
|---------|--------|----------|-----------|---------|------------|
| MNIST | Best Acc | **98.94** | 98.85 | 98.81 | 98.84 |
| | Final Acc | 98.72 | 98.63 | 98.58 | 98.83 |
| CIFAR-10 | Best Acc | **87.62** | 87.00 | 87.33 | 26.95 |
| | Final Acc | 86.69 | 86.63 | **87.33** | 17.89 |

As shown in Table 16, continuous score-function variants (Gaussian, Student-t, Laplace) produce stable training on both datasets. Gaussian noise provides the best or near-best results. Rademacher SPSA fails on CIFAR-10/ResNet-18 due to the high variance of two-point finite differences in complex settings, where the fixed-amplitude perturbation ($\pm 1.0$ per dimension) yields a coarse local approximation.

## J. Hybrid BP+LR Fusion Study

We investigate a hybrid gradient estimator that fuses BP and OVLR signals: $\hat{g} = \alpha \cdot \hat{g}_{\text{BP}} + (1 - \alpha) \cdot \hat{g}_{\text{OVLR}}$, where $\alpha = 0$ is pure OVLR and $\alpha = 1$ is pure BP. We conduct a 10-seed study on MNIST (4096 training samples, 20 epochs).

*Table 17.* Hybrid BP+LR fusion results on MNIST (10 seeds). Mean $\pm$ std of test accuracy (%) and training time (s).

| $\alpha$ | Accuracy (%) | Loss | Time (s) |
|----------|--------------|------|----------|
| 0.00 (pure OVLR) | $96.09 \pm 0.45$ | $0.207 \pm 0.032$ | $12.82 \pm 1.54$ |
| 0.25 | $96.16 \pm 0.43$ | $0.203 \pm 0.030$ | $11.52 \pm 1.34$ |
| 0.50 | $96.20 \pm 0.43$ | $0.198 \pm 0.035$ | $12.11 \pm 1.28$ |
| 0.75 | $96.15 \pm 0.49$ | $0.195 \pm 0.031$ | $11.34 \pm 0.79$ |
| 1.00 (pure BP) | $96.09 \pm 0.39$ | $0.188 \pm 0.031$ | $10.71 \pm 0.87$ |

As shown in Table 17, $\alpha$=0.50 achieves the best mean accuracy (96.20%), slightly above both pure OVLR (96.09%) and pure BP (96.09%). This suggests that a non-zero OVLR

contribution can complement BP's smoother gradient signal. However, the improvement is marginal, and pure BP remains the fastest option.

## K. True Black-Box Optimization: Full Results

We provide full per-problem results on the IOHexperimenter benchmarks introduced in Section 5.5.

### K.1. Discrete PBO Benchmarks

On discrete IOH PBO problems (dimension 32, budget 512, seeds 0/1/2), OVLR achieves the best aggregate performance:

*Table 18.* Discrete PBO benchmarks (dim=32, budget=512). Mean normalized score per problem.

| Method | OneMax | LeadOnes | Linear | IsingR | IsingT |
|--------|--------|----------|--------|--------|--------|
| OVLR | 1.000 | 0.958 | 1.000 | 1.000 | 1.000 |
| (1+1)-EA | 1.000 | 0.883 | 0.992 | 0.996 | 0.983 |
| CEM | 0.983 | 0.000 | 0.950 | 0.983 | 1.075 |
| REINFORCE | 0.963 | 0.000 | 0.942 | 0.588 | 1.325 |

OVLR achieves 100% success rate on IsingRing and the best score on the difficult LeadingOnes problem.

### K.2. Continuous BBOB Benchmarks

On continuous IOH BBOB objectives (dimension 10, budget 400, seeds 0/1/2):

*Table 19.* Continuous BBOB benchmarks (dim=10, budget=400). Mean normalized score.

| Method | Sphere | Ellipsoid | Rastrigin | Rosenbr. |
|--------|--------|-----------|-----------|----------|
| OVLR | 0.9129 | 0.0346 | 0.0017 | 0.0317 |
| CEM | 0.8433 | 0.0519 | 0.0000 | 0.0275 |
| (1+1)-ES | 0.2423 | 0.0082 | 0.0004 | 0.0212 |
| Gauss-REINF. | 0.0534 | 0.0012 | 0.0000 | 0.0178 |

OVLR achieves the highest normalized score on Sphere (0.9129) and Rosenbrock (0.0317), and remains competitive on Ellipsoid and Rastrigin. Applying OVLR to complex RL settings such as Atari/MuJoCo—where challenges such as high-dimensional state spaces, sparse rewards, and unstable dynamics arise—presents additional considerations beyond standard policy gradient methods, and is an important direction for future work.

## L. Details of Experimental Setup

### L.1. Datasets

We conduct experiments on three standard computer vision datasets with increasing complexity: CIFAR-10, CIFAR-100, and TinyImageNet. CIFAR-10 consists of 60,000 color

images of size $32 \times 32$ across 10 classes, while CIFAR-100 has the same format but includes 100 fine-grained classes. TinyImageNet is a subset of ImageNet containing 200 classes, with 500 training and 50 validation images per class at a resolution of $64 \times 64$ pixels.

For all datasets, we apply standard data augmentation including random horizontal flipping during training. Images are resized to $224 \times 224$ to match the input size requirements of Vision Transformer models. Dataset-specific normalization is applied using the following mean and standard deviation values: (CIFAR-10) $[0.4914, 0.4822, 0.4465]$ / $[0.2023, 0.1994, 0.2010]$; (CIFAR-100) $[0.5071, 0.4867, 0.4408]$ / $[0.2675, 0.2565, 0.2761]$; (TinyImageNet) $[0.4802, 0.4481, 0.3975]$ / $[0.2302, 0.2265, 0.2262]$.

### L.2. Network Architectures

We evaluate three representative architectures: ResNet-18, a residual network with 18 layers; DenseNet-121, a densely connected convolutional network with 121 layers; and ViT-B/16, a Vision Transformer using $16 \times 16$ patch size and pretrained on ImageNet-21K. The final classification heads are modified to match the number of classes in each dataset (10 for CIFAR-10, 100 for CIFAR-100, and 200 for Tiny-ImageNet).

### L.3. Training Methods

We compare standard backpropagation (BP) with OVLR, which employs symmetric noise with a scale of 1.0. The number of repeated samples is adjusted based on dataset complexity and is set to 100, 200, and 300 for CIFAR-10, CIFAR-100, and TinyImageNet, respectively, by default.

### L.4. Training Protocol

All models are trained for 30 or 50 epochs using the Adam optimizer with a batch size of 64. The learning rate is set to 0.001 for ResNet-18 and DenseNet-121, and reduced to 0.0001 for ViT-B/16 due to the use of pretrained weights. Training is performed on NVIDIA GPUs with CUDA support: except for the time and memory overhead tests—which are conducted on a single RTX 3090 to eliminate interference from other programs—all other experiments are run on an RTX 4090. For BP, we use the standard cross-entropy loss provided by PyTorch, whereas for OVLR, we employ the cross-entropy loss without reduction. The experiments are implemented using Python 3.9, with PyTorch 2.2.2 serving as the core deep learning framework.

