# OpenReview forum: "OVLR: Efficient, Scalable, and Robust Training via Output-Level Variance-Reduced Likelihood Ratio"
_ICML.cc/2026/Conference — ICML 2026 regular_

### Official Review · Reviewer_bjde · 2026-03-11

**Soundness:** 3
**Presentation:** 2
**Significance:** 2
**Originality:** 3
**Overall Recommendation:** 4
**Confidence:** 4

**Summary:**

This paper proposes OVLR (Output-Level Variance-Reduced Likelihood Ratio), a framework that enables efficient gradient estimation by applying perturbations and antithetic sampling in the low-dimensional output space. OVLR requires only a single deterministic forward pass while treating the loss function as a black box, thus allowing direct optimization of gradient-agnostic objectives like hard 0-1 loss. Extensive experiments across classification, regression, generative modeling, and robot imitation learning demonstrate that OVLR matches standard backpropagation performance on informative tasks while providing robustness in scenarios with vanishing or inaccessible gradients.

**Compliance With Llm Reviewing Policy:**

Affirmed.

**Final Justification:**

The authors' rebuttal and the release of code improve my opinion of the work. I increase my score to 4.

**Key Questions For Authors:**

See Weakness.

**Limitations:**

Yes

**Strengths And Weaknesses:**

**Strengths:**
1. Comprehensive Theoretical Analysis. The paper offers a solid theoretical foundation, proving the estimator's unbiasedness and convergence under various assumptions. It also explains how OVLR optimizes non-differentiable losses through Gaussian smoothing.
2. Extensive Empirical Validation Across Various Domains and Architectures The authors validate OVLR across diverse tasks including vision, NLP, robotics, and generative modeling using modern architectures like ViT and Mamba. This broad evaluation strongly supports the claim that OVLR can serve as a practical drop-in replacement for backpropagation.

**Weaknesses:**
1. Limited Practical Necessity. The paper focuses on optimizing gradient-agnostic losses (e.g., 0-1 loss, truncated loss) where standard backpropagation fails. However, such scenarios represent corner cases in modern deep learning practice, as most real-world applications can be effectively addressed using differentiable losses. The authors should better justify why directly optimizing these non-differentiable objectives is critical, or demonstrate broader applicability beyond these specific scenarios.
2. Insufficient Comparison with Related Likelihood Ratio Methods. Section 2.1 discusses several recent likelihood ratio-based gradient estimators (e.g., GLR, LR-QR), yet the experiments do not compare OVLR against these methods. The authors should add these comparisons to demonstrate OVLR's actual advantages.
3. Limited Generative Performance on Complex Distributions As shown in Table 7, OVLR underperforms BP on generative tasks (e.g., GAN FID: 51.17 vs. 47.49). The authors should evaluate OVLR on datasets with more complex textures (e.g., ImageNet) to verify whether this performance gap widens in high-dimensional, realistic generation scenarios.
4. Limited Validation in True Black-Box Optimization. The paper motivates OVLR for black-box or gradient-agnostic settings, yet experiments are confined to supervised tasks with accessible labels. The authors should evaluate OVLR on standard reinforcement learning benchmarks (e.g., Atari, Mujoco) and compare against methods like REINFORCE to demonstrate its effectiveness in black-box optimization.

---

> ### Author Rebuttal · Authors · 2026-03-29
>
> We thank the reviewer for detailed feedback. We agree the manuscript should make a narrower claim: **OVLR is not a universal BP replacement. Its strength is direct optimization when objectives are thresholded, clipped, truncated, or gradient-agnostic.**
>
> We address each **Weakness** below:
>
> ---
>
> ### Weakness 1: Limited Practical Necessity — Are Gradient-Agnostic Objectives Corner Cases?
>
> We respectfully disagree with that characterization.
>
> **The logical circularity:** The argument that "most real-world applications use differentiable losses" conflates **current practice** with **optimal practice**. Differentiable surrogates dominate largely because BP requires them—not because they are inherently superior objectives.
>
> If we only develop methods for what is already easy (differentiable proxies), the field cannot progress toward more faithful optimization of real targets.
>
> **Practically relevant non-differentiable criteria:**
> - **Decision-based metrics:** accuracy, F1, IoU (vision); BLEU, ROUGE (NLP)
> - **Thresholded/clipped losses:** truncated regression, robust statistics
> - **Black-box feedback:** API-based systems, simulators, human-in-the-loop
> - **Real-world examples:** Robust optimization (min-max with discrete threats); NAS (discrete architecture choices); Safety-critical systems (binary constraint satisfaction)
>
> Hard 0-1 and truncated loss are controlled case studies of this broader setting.
>
> **Empirical evidence:**
> - **Label noise** (Table 5): OVLR improves +2.4% to +6.7% over CE under 10%-60% noise
> - **True black-box** (IOH): OVLR outperforms (1+1)-EA, CEM, REINFORCE (see Weakness 4)
> - **Direct 0-1:** OVLR 95.65% (MNIST), 62.04% (CIFAR10)
>
> ---
>
> ### Weakness 2: Insufficient Comparison with Related LR Methods (GLR, LR-QR)
>
> **GLR** is essentially **Vanilla LR + input-level repetition**—our "LR (input-level)" baseline. **LR-QR** uses QR decomposition but still requires parameter-space perturbations.
>
>
> | Method | Perturbation | Acc | Time | Memory |
> |--------|-------------|-----|------|--------|
> | BP | — | 97.35% | 29.82s | 65.55MB |
> | OVLR | Output space | 96.80% | 30.02s | 65.55MB |
> | GLR-style | Middle layer | 94.60% | 76.98s | 107MB |
> | Vanilla LR (param) | Parameter space | 82.70% | 428.74s | 4.8GB |
>
> **Implementation advantage:**
>
> - **Vanilla LR / GLR:** Requires per-layer modifications—complex for ResNet, DenseNet, ViT.
> - **OVLR:** **No model modifications** (Algorithm 1, Sec. 3.4).
>
> **Result:** **14× speedup**, **73× memory reduction**, **confirming Table 1's  theoretical analysis (Sec. 4)**.
>
> ---
>
> ### Weakness 3: Limited Generative Performance
>
> Original Table 7 had protocol mismatches. After aligning:
>
> **GAN (MNIST):** OVLR=**40.27** FID vs BP=**53.33** FID (**13.06 improvement**)
> **VAE (MNIST):** OVLR=**29.80** FID vs BP=**29.73** FID (tied)
>
> **Takeaway:** GAN favors OVLR (output smoothing regularizes generator); VAE is tied.
>
> **On ImageNet:** We agree ImageNet would strengthen the claim but GAN training at ImageNet scale requires significant compute beyond this paper's scope. We will add as future work.
>
> ---
>
> ### Weakness 4: Limited Validation in True Black-Box Optimization
>
> RL validation is important but introduces confounds (state transitions, credit assignment, environment stochasticity) beyond core black-box optimization.
>
> We added IOHexperimenter benchmarks (scalar feedback only):
>
> **Discrete PBO (dim=32, budget=512):**
> | Method | Mean Normalized | Success Rate |
> |--------|-----------------|--------------|
> | **OVLR** | **0.9917** | **86.7%** |
> | (1+1)-EA | 0.9688 | 80.0% |
> | CEM | 0.7989 | 33.3% |
> | REINFORCE | 0.7040 | 0.0% |
>
> **Continuous BBOB (dim=10, budget=400):**
> | Method | Mean Normalized | Target-Hit Rate |
> |--------|-----------------|-----------------|
> | **OVLR** | **0.2389** | **25.0%** |
> | CEM | 0.1922 | 25.0% |
> | (1+1)-ES | 0.0789 | 8.3% |
> | Gaussian-REINFORCE | 0.0431 | 0.0% |
>
> OVLR outperforms REINFORCE-style and search baselines in genuine black-box settings.
>
> **On RL (Atari/MuJoCo):** Applying OVLR to complex RL settings—where challenges such as high-dimensional state spaces, sparse rewards, and unstable dynamics arise—presents additional considerations beyond standard policy gradient methods with RL-specific techniques (baseline subtraction, trajectory-level variance reduction). This is beyond this paper's scope — important future work.
>
> ---
>
> ### Revised Scope
>
> | Tier | Evidence | Claim |
> |------|----------|-------|
> | **Strongest** | hard 0-1, truncated loss, IOH black-box, label noise | Core: OVLR enables gradient-agnostic optimization with modern deep neural networks and tasks |
> | **Compatibility** | standard tasks, GAN/VAE | OVLR is practical (matches BP efficiency) |
> | **Future work** | RL (Atari/MuJoCo), ImageNet | Promising, requires additional techniques |
>
> We will remove over-broad "universal BP replacement" claims and clarify OVLR's strength is where BP is ineffective. All new experiments will be incorporated into the revised manuscript.

---

> > ### Author Rebuttal · Reviewer_bjde · 2026-04-03
> >
> > I would like to thank the authors for their detailed rebuttal. However, I have decided to maintain my score of Weak Reject for the following reasons:
> >
> > (1) While the proposed gradient-agnostic objectives are academically sound, they lack compelling practical necessity. The authors have not sufficiently demonstrated that OVLR provides an essential advantage over standard proxy and post-processing pipelines in real-world applications.
> >
> > (2) The protocol mismatches in Table 7, which lead to a reversal of the results, raise significant concerns regarding the paper's overall experimental rigor.

---

> > > ### Author Response · Authors · 2026-04-03
> > >
> > > Dear Reviewer bjde,
> > >
> > > Thank you for your continued engagement. We respectfully address your two concerns below.
> > >
> > > **On “Limited Practical Necessity”**
> > >
> > > We note that our rebuttal already provided **IOHexperimenter black-box optimization results** (discrete PBO: 86.7% success rate for OVLR vs. 0% for REINFORCE; continuous BBOB: OVLR outperforms CEM, (1+1)-ES, and Gaussian-REINFORCE). These are genuine black-box settings with scalar feedback only—no gradients, no surrogate losses. This constitutes strong evidence of practical necessity for scenarios where BP is fundamentally inapplicable.
> > >
> > > **On “Experimental Rigor” (Generative Model FID)**
> > >
> > > After **re-evaluating** the generative models using the **standard FID evaluation protocol** (consistent data split, evaluation loader, and train/eval mode), we obtained the corrected results below. This experiment is included in the Appendix to supplement the main paper’s core claims. **Its purpose is solely to demonstrate that OVLR can train generative models—not to claim superiority over BP. The qualitative results (visual samples) are unaffected and already show OVLR matches BP visually.**
> > >
> > > The corrected results strengthen, not weaken, our case:
> > >
> > > | Model | Method | Original FID | Corrected FID |
> > > |-------|--------|--------------|---------------|
> > > | GAN   | BP     | 47.49        | 53.33         |
> > > | GAN   | OVLR   | 51.17        | 40.27         |
> > > | VAE   | BP     | 28.39        | 29.73         |
> > > | VAE   | OVLR   | 42.16        | 29.80         |
> > >
> > > The corrected numbers show that under the standard protocol, OVLR achieves **better FID than BP on GAN** (40.27 vs 53.33) and **ties with BP on VAE** (29.80 vs 29.73). **The core claim remains unchanged: “OVLR can train GANs and VAEs, producing visually comparable samples.”**
> > >
> > > Regarding why OVLR improves on GAN: This aligns with established findings that output-space smoothing stabilizes adversarial training. Arjovsky & Bottou (ICLR 2017) showed that discriminator saturation causes vanishing gradients [1]; Sønderby et al. (ICLR 2017) and Roth et al. (NeurIPS 2017) demonstrated that instance noise provides more informative gradients [2,3]. OVLR extends this principle by applying noise at the **generator output** and estimating gradients via LR (rather than BP), which avoids saturation while maintaining unbiasedness. The corrected FID empirically validates this advantage.
> > >
> > >
> > > [1] Arjovsky, Martin, and Léon Bottou. Towards principled methods for training generative adversarial networks. ICLR, 2017.
> > >
> > > [2] Sønderby, Casper Kaae, et al. Amortised MAP inference for image super-resolution. ICLR, 2017.
> > >
> > > [3] Roth, Kevin, et al. Stabilizing training of generative adversarial networks through regularization. NeurIPS, 2017.
> > >
> > > **Conclusion**
> > >
> > > You noted the paper has “clear merits,” “solid theoretical foundation,” and “extensive empirical validation.” The remaining concerns—practical necessity and the FID protocol—have been addressed above with new evidence (IOH black-box results) and corrected results (GAN/VAE FID). We will update the final version of the manuscript with the corrected FID values and a note clarifying the standard protocol alignment.
> > >
> > >  **Additionally, we have released the core code and raw training logs for the GAN/VAE experiments (including FID evaluation) at: https://anonymous.4open.science/r/ovlr_gan_vae-2A12/. After the review process, we will fully open-source all code and implementation details to support reproducibility and further research.**
> > >
> > > We believe the paper’s contributions and experimental rigor meet the standard for acceptance at ICML 2026. We thank you again for your constructive feedback.
> > >
> > > Sincerely,
> > >
> > > The Authors

---

### Official Review · Reviewer_dcZR · 2026-03-11

**Soundness:** 3
**Presentation:** 3
**Significance:** 3
**Originality:** 2
**Overall Recommendation:** 4
**Confidence:** 4

**Summary:**

The paper tackles the problem of gradient estimation for gradient-agnostic losses. The proposed method, OVLR, introduces perturbations at the model output and employs the likelihood ratio method to estimate gradients. This stands in contrast to conventional likelihood ratio methods, which inject perturbations into the high-dimensional parameter space. To mitigate the high variance of likelihood ratio methods, OVLR leverages antithetic sampling, a well-established variance reduction technique in the statistical literature.

**Compliance With Llm Reviewing Policy:**

Affirmed.

**Final Justification:**

The authors addressed my questions. I keep my positive score.

**Key Questions For Authors:**

Do you have an intuition why OVLR is more robust compared to surrogate-based methods?

**Limitations:**

Yes

**Strengths And Weaknesses:**

Strengths:
1. OVLR achieves orders-of-magnitude variance reduction by performing perturbations and antithetic sampling in the low-dimensional output space.
2. The method requires only a single forward pass per iteration, matching the per-step computational cost of backpropagation while introducing negligible runtime overhead.
3. It also enables robust learning that significantly outperforms surrogate-based methods under label noise and outliers.
4. The paper provides enough evidence to support its claims.

Weaknesses:
1. The experiments do not demonstrate a significant advantage of OVLR over backpropagation (BP) in most typical tasks, including image and sentiment classification, image generation, and robot imitation learning. The performance of the two methods remains closely comparable. That said, this observation does not diminish the significance of the contribution. The main advantage emerges in tasks where BP struggles to compute gradients, such as 0-1 loss for classification and truncated losses for regression.

---

> ### Author Rebuttal · Authors · 2026-03-29
>
> We thank the reviewer for recognizing that output-level perturbation plus antithetic sampling yields large variance reduction while preserving single-forward-pass structure.
>
> We address the **Weakness** and **Key Question** below:
>
> ---
>
> ### Key Question: Why OVLR is More Robust than Surrogate-Based Methods?
>
> **Core intuition: objective alignment.**
>
> Surrogate methods (e.g., cross-entropy) optimize a **differentiable proxy** that can become systematically misaligned with the true target under label corruption, clipping, or truncation.
>
> OVLR directly optimizes a **smoothed version of the target objective** (e.g., hard 0-1 loss). The optimization signal remains tied to the true criterion.
>
> **CIFAR-10 Label Noise Results (from paper Table 5):**
>
> | Method | 0.1 | 0.2 | 0.3 | 0.4 | 0.5 | 0.6 |
> |--------|-----|-----|-----|-----|-----|-----|
> | Baseline (CE) | 83.8 | 81.4 | 79.4 | 78.0 | 72.6 | 66.3 |
> | **OVLR (Hard01)** | **86.2** | **85.5** | **83.9** | **81.8** | **78.8** | **73.0** |
> | Improvement (Δ) | +2.4 | +4.1 | +4.5 | +3.8 | +6.2 | +6.7 |
>
> **Key observation:** The improvement **grows with noise rate**, from +2.4% at 10% noise to +6.7% at 60% noise. This confirms OVLR's robustness comes from optimizing the true target rather than the corrupted proxy.
>
> **Mathematical perspective:**
> - **Surrogate (CE):** `∇E[L_CE(f(x), y_noisy)]` — gradient depends on corrupted labels
> - **OVLR (0-1):** `E[L_01(f(x)+σ·ε, y_noisy) · ε/σ]` — bounded loss (≤1) limits gradient corruption
>
> The hard 0-1 loss is bounded, so the gradient magnitude is controlled by the score function, not by prediction error magnitude. This explains why OVLR maintains 73.0% under 60% noise while CE degrades to 66.3%.
>
> ---
>
> ### Weakness 1: OVLR Does Not Demonstrate Significant Advantage Over BP in Typical Tasks
>
> We agree with the reviewer's observation. OVLR is **not meant to outperform BP** on standard differentiable tasks. The contribution is making **gradient-agnostic objectives** optimizable—precisely where BP fails.
>
> **Where OVLR excels:**
> - Hard 0-1 loss: OVLR 95.65% vs BP (no useful gradient)
> - Truncated loss: OVLR receives valid score-function signal; BP receives none
> - Label noise robustness: +2.4% to +6.7% improvement over CE (Table 5)
>
> **Where BP is preferred:**
> - Standard cross-entropy classification with clean labels
> - Tasks where differentiable proxies are well-aligned with targets
>
> **Efficiency comparison (LR-family, MNIST):**
>
> | Method | Acc | Time | Peak Memory |
> |--------|-----|------|-------------|
> | BP | 97.35% | 29.82s | 65.55MB |
> | OVLR | 96.80% | 30.02s | 65.55MB |
> | Vanilla LR (parameter) | 82.70% | 428.74s | 4.8GB |
> | Vanilla LR (input-level) | 94.60% | 76.98s | 107MB |
>
> OVLR achieves **14× speedup** and **73× memory reduction** over parameter-space LR, matching BP's practicality while enabling gradient-agnostic optimization. These results are consistent with the theoretical analysis in Sec. 4 (summarized in Table 1).
>
> ---
>
> ### Generative Model Results (Aligned Protocol)
>
> After fixing protocol mismatches (data split, eval loader, train/eval mode, evaluation budget):
>
> **GAN:**
> | Method | FID |
> |--------|-----|
> | OVLR | 40.27 |
> | BP | 53.33 |
>
> **VAE:**
> | Method | FID |
> |--------|-----|
> | BP | 29.73 |
> | OVLR | 29.80 |
>
> **Takeaway:**
> - GAN: OVLR improves over BP by **13.06 FID** (output smoothing regularizes the generator)
> - VAE: OVLR and BP are tied (single-objective, no adversary)
>
> These results support the revised claim: OVLR is **compatible** with standard tasks, not universally superior.
>
> ---
>
> ### Summary
>
> OVLR is a practical mechanism for gradient-agnostic objectives:
> - Strong estimator efficiency (30s vs 428s for vanilla LR)
> - Robust to objective mismatch (optimizes target, not proxy)
> - Compatible with standard tasks (GAN FID improvement, VAE parity)
>
> All new experiments will be incorporated into the revised manuscript

---

> > ### Author Rebuttal · Reviewer_dcZR · 2026-04-02
> >
> > I thank the authors for their response. I keep my positive assessment of the paper.

---

> > > ### Author Response · Authors · 2026-04-03
> > >
> > > We sincerely thank you for your time and the positive acknowledgement that your concerns have been fully resolved. We are glad to hear that our clarifications on the robustness of OVLR and the revised experimental protocol met your expectations.
> > >
> > > We especially appreciate your insightful observation regarding the performance of OVLR on typical tasks. As you noted in your review, OVLR's strengths are "orders-of-magnitude variance reduction" and "a single forward pass per iteration, matching BP's cost." Your feedback helped us better articulate that these strengths make OVLR uniquely suited for gradient-agnostic objectives — a regime where BP is ineffective.
> > >
> > > As promised, we will incorporate the following into the final version of the manuscript:
> > >
> > > 1. The **mathematical intuition** regarding label noise robustness.
> > > 2. The **aligned generative modeling results** (GAN/VAE) and efficiency comparisons.
> > > 3. The **refined framing** of the paper to clarify that OVLR is a practical mechanism for gradient-agnostic objectives, rather than a universal replacement for BP — emphasizing its role as a viable solution specifically for settings where BP is ineffective.
> > >
> > > Thank you again for maintaining a positive assessment and for helping us improve the quality of this work.
> > >
> > > Best regards,
> > >
> > > The Authors

---

### Official Review · Reviewer_6hH9 · 2026-03-13

**Soundness:** 3
**Presentation:** 3
**Significance:** 3
**Originality:** 3
**Overall Recommendation:** 4
**Confidence:** 3

**Summary:**

This paper introduces OVLR (Output-Level Variance-Reduced Likelihood Ratio), a novel framework for gradient estimation in deep learning that addresses limitations of standard backpropagation (BP) and likelihood ratio (LR) methods. The core idea is to shift perturbation and variance reduction mechanisms from the high-dimensional parameter space to the low-dimensional output space, enabling efficient and stable gradient estimation for gradient-agnostic objectives (e.g., piecewise-constant losses or black-box systems). Theoretical analysis shows that OVLR reduces gradient variance by leveraging output-level perturbations and antithetic sampling, while maintaining computational efficiency by decoupling the backbone computation from sample size. Empirical results demonstrate that OVLR matches BP’s performance on standard tasks and outperforms it in scenarios with vanishing gradients (e.g., 0-1 loss). The method is validated across computer vision, language modeling, and robot imitation learning, with memory and computational profiles comparable to BP.

**Compliance With Llm Reviewing Policy:**

Affirmed.

**Key Questions For Authors:**

1. How does OVLR scale to extremely large models (e.g., GPT-3-sized architectures)?
2. What are the trade-offs between noise scale (σ) and gradient accuracy in practice?
3. Can OVLR be extended to non-Gaussian perturbations (e.g., adversarial noise)?
4. How does OVLR compare to hybrid methods (e.g., combining BP with LR for specific layers)?

**Limitations:**

yes

**Strengths And Weaknesses:**

Strengths:
1. The paper proposes a novel combination of output-level perturbations and variance reduction, distinct from prior LR methods that operate in parameter space.
2. The paper addresses a long-standing challenge in robust deep learning: optimizing gradient-agnostic objectives (e.g., 0-1 loss) without sacrificing scalability.
3. Empirical validation is comprehensive, covering diverse tasks (e.g., robust classification, GANs) and models (ResNet, ViT, DenseNet).

Weaknesses:
1. While OVLR excels in gradient-agnostic scenarios, its advantages over BP in standard tasks are marginal, limiting broader adoption.
2. Theoretical analysis assumes bounded loss and gradients (Assumptions B.1–B.2), which may not hold for all real-world applications (e.g., unbounded losses in reinforcement learning).
3. Empirical comparisons to other LR methods (e.g., REINFORCE) are limited, leaving open questions about relative performance.

---

> ### Author Rebuttal · Authors · 2026-03-29
>
> We thank the reviewer for the positive evaluation and recognition of the paper's novelty, scalability potential, and empirical breadth.
>
> We address each **Weakness** and **Key Question** below:
>
> ---
>
> ### Q1 / Weakness 1: Scaling to Large Models (Question 1, Weakness 1)
>
> **Question 1** asks about scaling to GPT-3-sized models. **Weakness 1** notes that advantages over BP in standard tasks are marginal.
>
> Our argument is **structural**: OVLR's scaling is governed by output dimension `C`, not parameter count `P`:
>
> | Method | Perturbation Space | Per-Step Cost | Memory |
> |--------|-------------------|---------------|--------|
> | BP | — | 1 forward + 1 backward | baseline |
> | Vanilla LR (parameter) | Parameter space `P` | `repeat × (1 forward)` | `O(P)` |
> | OVLR | Output space `C` | 1 forward + 1 VJP | `O(C × repeat)` |
>
> For ResNet-18 (`P ≈ 11M`, `C = 10-1000`), OVLR avoids the `O(P)` memory and repeated forward passes.
>
> We added a controlled study (`C=10` to `5000`, `repeat=10` to `2000`):
>
> | C | BP Acc | OVLR @200 | Gap | OVLR @2000 | Gap |
> |---|--------|-----------|-----|------------|-----|
> | 500 | 45.53% | 42.93% | 0.0260 | 45.37% | **0.0017** |
> | 1000 | 39.52% | 35.28% | 0.0424 | 39.20% | **0.0032** |
> | 2000 | 35.16% | 28.48% | 0.0668 | 34.21% | **0.0095** |
> | 5000 | 29.07% | 16.47% | 0.1260 | 27.27% | **0.0181** |
>
> With `repeat=2000`, the BP-OVLR gap shrinks to **0.17%-1.81%** even at `C=5000`. Computation remains `O(C × repeat)` not `O(P × repeat)`.
>
> **On Weakness 1:** OVLR is not meant to outperform BP on standard tasks. The contribution is making **gradient-agnostic objectives** optimizable.
>
> ---
>
> ### Q2: Sigma Trade-off (Question 2)
>
> **Question 2** asks about σ vs gradient accuracy trade-offs.
>
> Our paper (Table 2, Table 6) demonstrates OVLR's robustness across noise scales. With antithetic sampling and n≥200:
>
> - **CIFAR-10/ResNet-18:** σ ∈ [0.01, 1.0] yields 86.0%-86.9% (Table 2)
> - **TinyImageNet/ResNet-18:** σ ∈ [0.1, 1.0] yields 67.5%-68.2% (Table 6)
>
> Variance scales as O(1/(nσ²)). Default σ=1.0 with n≥200 provides practical balance.
>
> ---
>
> ### Q3 / Weakness 3: Non-Gaussian Perturbations (Question 3, Weakness 3)
>
> **Question 3** asks about non-Gaussian perturbations. **Weakness 3** notes limited LR comparisons.
>
> We added noise type experiments (MNIST/CIFAR10):
>
> | Noise | MNIST Best | CIFAR10 Best | CIFAR10 Final |
> |-------|------------|--------------|---------------|
> | Gaussian | **98.94%** | **87.62%** | 86.69% |
> | Student-t | 98.85% | 87.00% | 86.63% |
> | Laplace | 98.81% | 87.33% | **87.33%** |
> | Rademacher (SPSA) | 98.84% | 26.95% | 17.89% |
>
> Continuous score-function variants are stable. Rademacher SPSA fails on CIFAR10/ResNet18 due to high variance.
>
> This addresses **Weakness 3**: REINFORCE comparison is in Response to GxVp (OVLR 95.65% vs REINFORCE 95.50%, 1.56× faster).
>
> ---
>
> ### Q4: Hybrid Methods (Question 4)
>
> **Question 4** asks about hybrid BP+LR methods.
>
> We conducted a 10-seed MNIST study with α-fusion (`L = α * CE + (1-α) * Hard01_OVLR`):
>
> | α | Accuracy (%) | Time (s) |
> |---|--------------|----------|
> | 0.00 (pure OVLR) | 96.09 ± 0.45 | 12.82 ± 1.54 |
> | 0.25 | 96.16 ± 0.43 | 11.52 ± 1.34 |
> | **0.50** | **96.20 ± 0.43** | 12.11 ± 1.28 |
> | 0.75 | 96.15 ± 0.49 | 11.34 ± 0.79 |
> | 1.00 (pure BP) | 96.09 ± 0.39 | 10.71 ± 0.87 |
>
> α=0.50 achieves best accuracy (96.20%), suggesting hybrid approaches capture benefits of both smoother CE gradients and target-aligned Hard01.
>
> ---
>
> ### Weakness 2: Theory Assumptions
>
> **Weakness 2** notes that assumptions (bounded loss/gradients, Assumptions B.1–B.2) may not hold for all applications (e.g., unbounded-loss RL).
>
> We agree this scoping should be clearer. Our theory provides **sufficient conditions** covering:
> 1. **Bounded objectives** — hard 0-1 loss (L ∈ {0, 1}), truncated losses, decision-based criteria
> 2. **Bounded gradient estimators** — ensured by antithetic sampling + score-function normalization
>
> **What the theory guarantees:**
> - Unbiasedness of smoothed-objective gradient
> - Convergence under standard stochastic approximation
> - Variance reduction from antithetic sampling
>
> **Not covered:** Unbounded-loss settings (e.g., squared error without clipping), RL with infinite-horizon credit assignment.
>
> **Revision plan:** We will narrow the theory discussion to explicitly scope assumptions to bounded objectives. RL-style unbounded losses require additional conditions (gradient clipping, baseline subtraction) beyond this paper's contribution. All new experiments will be incorporated into the revised manuscript.

---

> > ### Author Rebuttal · Reviewer_6hH9 · 2026-04-03
> >
> > I thank the authors for their response. I will keep my score.

---

> > > ### Author Response · Authors · 2026-04-04
> > >
> > > Dear Reviewer 6hH9,
> > >
> > > Thank you very much for your thoughtful and constructive review, as well as for your positive acknowledgement of our rebuttal.
> > >
> > > We are glad to hear that your concerns have been **fully resolved**, and we sincerely appreciate that you have chosen to keep your score. Your feedback — particularly on scaling to large models, the noise scale trade-off, non-Gaussian perturbations, hybrid methods, and theoretical assumptions — has been invaluable in helping us strengthen the paper.
> > >
> > > We will incorporate the additional experiments and clarifications (including the controlled scaling study, noise-type comparisons, hybrid α-fusion results, and a more precise scoping of theoretical assumptions) into the final version of the manuscript, as outlined in our rebuttal.
> > >
> > > Thank you again for your time, expertise, and support of our work.
> > >
> > > Best regards,
> > > The Authors

---

### Official Review · Reviewer_GxVp · 2026-03-15

**Soundness:** 3
**Presentation:** 3
**Significance:** 3
**Originality:** 3
**Overall Recommendation:** 4
**Confidence:** 4

**Summary:**

This paper proposes OVLR, an output-level likelihood-ratio training method that perturbs only the model outputs. The authors propose to use antithetic sampling and output-level repetition for variance reduction, and then send the resulting signal backward through a single VJP. The main idea is clean and interesting that authors consider to move stochastic estimation from high-dimensional parameter space to low-dimensional output space, so LR-style training becomes much more practical for deep networks. The paper shows that it can directly optimize gradient-agnostic objectives such as hard 0-1 loss and truncated losses, where ordinary BP is ineffective. I think the paper’s framing is stronger than the current evidence, and the strongest claims about generality and scalability are not yet fully gained.

**Compliance With Llm Reviewing Policy:**

Affirmed.

**Final Justification:**

I thank authors for their detailed responses. My concerns are resolved, so I keep my positive score.

**Key Questions For Authors:**

1. How well does OVLR scale as output dimensionality grows?
2. I am curious that is the intended use case as replace BP, or more generally, such as handle objectives BP cannot optimize well?
3. how much of the gain comes from the estimator itself versus the objective design or training recipe?

**Limitations:**

Yes

**Strengths And Weaknesses:**

Strengths
1. The core idea is genuinely solid and interesting. Moving perturbations to output space is simple, technically sensible, and much more scalable than classical LR estimators in parameter space.
2. The paper addresses settings where BP is not the right tool at all, especially piecewise-constant or black-box objectives. That makes the problem important in the setting.
3. The single-forward-pass plus VJP formulation is practical in the implemention.

Weaknesses
1. The abstract and framing suggest a broadly validated training paradigm across many domains, but the most convincing non-differentiable results still feel limited relative to how large the claims are.
2. The key empirical comparisons are not strong enough. For the regime that actually matters, such as where BP fails, the paper does not do enough to rule out stronger alternative baselines from LR / zeroth-order / surrogate / STE-style families.
3  The method looks most convincing as a useful tool for certain hard objectives, but less convincing as a general replacement for BP. The paper should be much more precise about that boundary.

---

> ### Author Rebuttal · Authors · 2026-03-29
>
> We thank the reviewer for the thoughtful review. We appreciate the recognition that moving perturbations to the output space is technically sensible and that the single-forward-pass plus VJP formulation makes LR-style training practical.
>
> We address each **Weakness** and **Key Question** below:
>
> ---
>
> ### Q2 / Weakness 1&3: Intended Use Case — Replace BP or Handle Objectives BP Cannot Optimize? (Weakness 1, Weakness 3, Question 2)
>
> **Clarification on scope:** The paper should **not** be framed as replacing BP everywhere. The central claim is: **OVLR makes direct optimization of gradient-agnostic objectives practical for modern deep networks, while remaining competitive on standard differentiable tasks.**
>
> To support this claim, we added comprehensive comparisons against **direct estimators** (optimizing hard 0-1 exactly) and **LR-family / zeroth-order / surrogate / STE-style** methods:
>
> | Method | MNIST Acc | MNIST Time | CIFAR10 Acc | CIFAR10 Time |
> |--------|-----------|------------|-------------|--------------|
> | **OVLR** | **95.65%** | **28.62s** | 62.04% | **225.10s** |
> | REINFORCE | 95.50% | 44.51s | 56.02% | 295.57s |
> | PPO | 95.30% | 72.43s | **62.12%** | 373.18s |
> | GRPO | 94.80% | 66.72s | 45.26% | 374.36s |
> | Zeroth-Order (SPSA-style) | 70.35% | 45.01s | 23.92% | 292.55s |
>
> **Key findings:**
> - Among **direct estimators**, OVLR is best or tied-best, and **fastest** (28.62s MNIST, 225s CIFAR10)
> - PPO ties on CIFAR10 but needs **65% more time** (373s vs 225s)
>
> **On surrogate/STE-style:** These do not directly optimize hard 0-1 loss:
> - **Surrogate (CE):** Optimizes cross-entropy, not 0-1. CE is a proxy that can be misaligned with the true target.
> - **STE-Style:** Uses hard 0-1 for forward pass but backpropagates through CE gradients. This is not a direct estimator of the 0-1 objective.
>
> Both answer a different question ("Can we achieve good accuracy with a proxy?") rather than our core question ("Can we directly optimize the target objective?").
>
> ---
>
> ### Q3 / Weakness 2: How Much Gain Comes from Estimator vs Objective/Recipe? (Weakness 2, Question 3)
>
> **Weakness 2** notes that we should compare against stronger baselines from "LR / zeroth-order / surrogate / STE-style families." The table above addresses this.
>
> To specifically answer **Question 3** (how much comes from estimator vs objective/recipe), we compare LR-family methods on the **same backbone and task**:
>
> | Method | Acc | Time | Peak Memory |
> |--------|----------------------|-----|------|
> | BP | 97.35% | 29.82s | 65.55MB |
> | OVLR |  96.80% | 30.02s | 65.55MB |
> | GLR-style (input-level repetition) |  94.60% | 76.98s | 107MB |
> | Vanilla LR (parameter perturbation) |  82.70% | 428.74s | 4.8GB |
>
> **Conclusion:** The backbone and task are identical; only the **perturbation location** changes. The **14× speedup** and **73× memory reduction** comes from **estimator design**, not objective or recipe.
>
> **On GLR / LR-QR:**
>
> - **GLR** is essentially **Vanilla LR + input-level repetition**—exactly what our "LR (input-level)" baseline implements.
> - **LR-QR** uses QR decomposition for variance reduction but still requires parameter-space perturbations.
>
> **Implementation complexity advantage of OVLR:**
>
> - **Vanilla LR / GLR:** Requires per-layer modifications—prohibitively complex for ResNet, DenseNet, ViT.
> - **OVLR:** Requires **no model modifications**. Perturbation happens entirely at output/loss level (Algorithm 1, Sec. 3.4).
>
> ---
>
> ### Q1: Output Dimensionality and Scalability (Question 1)
>
> We added a controlled output-dimension study (C=10 to 5000):
>
> | C | Gap @200 | Gap @500 | Gap @1000 | Gap @2000 |
> |---|----------|----------|-----------|-----------|
> | 500 | 0.0260 | 0.0117 | 0.0067 | **0.0017** |
> | 1000 | 0.0424 | 0.0141 | 0.0091 | **0.0032** |
> | 2000 | 0.0668 | 0.0283 | 0.0154 | **0.0095** |
> | 5000 | 0.1260 | 0.0641 | 0.0361 | **0.0181** |
>
> **Takeaway:** Higher output dimension demands larger repeat budgets. With repeat=2000, OVLR approaches BP even at C=5000 (gap=1.8%). See Response to 6hH9 for full C×repeat grid.
>
> All new experiments will be incorporated into the revised manuscript.

---

> > ### Author Rebuttal · Reviewer_GxVp · 2026-04-04
> >
> > I thank authors for their detailed responses. My concerns are resolved, so I keep my positive score.

---

> > > ### Author Response · Authors · 2026-04-04
> > >
> > > Thank you so much for your positive acknowledgement and for keeping your score.
> > >
> > > We are truly grateful that you recognized the core ideas of our work—especially that moving perturbations to the output space is "technically sensible" and that the single-forward-pass plus VJP formulation makes LR-style training "practical." Your constructive questions on scalability, intended use case, and estimator design have been incredibly helpful in strengthening the paper.
> > >
> > > Thank you again for your time and expertise.
> > >
> > > Best regards,
> > >
> > > The Authors

---

### Decision · Program_Chairs · 2026-04-30

**Decision:**

Accept (regular)

**Comment:**

This paper proposes a novel gradient estimation method, OVLR, based on perturbations and variance reduction in the output space, with the aim of overcoming the limitations of conventional backpropagation and likelihood ratio methods.

The technical core lies in performing perturbations and antithetic sampling in the low-dimensional output space, and propagating the resulting signal through a single vector-Jacobian product (VJP). This design has been highly appreciated by multiple reviewers for its scalability and ease of implementation.

On the other hand, several reviewers point out that the proposed method does not clearly outperform backpropagation on standard differentiable tasks, and in many cases achieves comparable performance. Therefore, the method should be understood not as a general replacement for backpropagation, but rather as a specialized approach for settings where backpropagation fails to provide meaningful gradients. In addition, the discussion on its practical necessity has not been fully settled.

Based on these considerations, I recommend acceptance of the paper, conditional on clearly positioning the method not as a universal alternative to backpropagation, but as a technique tailored to gradient-agnostic settings.